# Role of network-mediated stochasticity in mammalian drug resistance

Kevin S. Farquhar [1,2], Daniel A. Charlebois [1,6], Mariola Szenk[1,3], Joseph Cohen[3], Dmitry Nevozhay[4,5] & Gábor Balázsi[1,2,3,5]

A major challenge in biology is that genetically identical cells in the same environment can display gene expression stochasticity (noise), which contributes to bet-hedging, drug tolerance, and cell-fate switching. The magnitude and timescales of stochastic fluctuations can depend on the gene regulatory network. Currently, it is unclear how gene expression noise of specific networks impacts the evolution of drug resistance in mammalian cells. Answering this question requires adjusting network noise independently from mean expression. Here, we develop positive and negative feedback-based synthetic gene circuits to decouple noise from the mean for Puromycin resistance gene expression in Chinese Hamster Ovary cells. In low Puromycin concentrations, the high-noise, positive-feedback network delays long-term adaptation, whereas it facilitates adaptation under high Puromycin concentration. Accordingly, the low-noise, negative-feedback circuit can maintain resistance by acquiring mutations while the positive-feedback circuit remains mutation-free and regains drug sensitivity. These findings may have profound implications for chemotherapeutic inefficiency and cancer relapse.

[1] The Louis and Beatrice Laufer Center for Physical and Quantitative Biology, Stony Brook University, Stony Brook, NY 11794, USA. [2] Genetics and Epigenetics Graduate Program, The University of Texas MD Anderson Cancer Center, UT Health Graduate School of Biomedical Sciences, Houston, TX 77030, USA. [3] Department of Biomedical Engineering, Stony Brook University, Stony Brook, NY 11794, USA. [4] School of Biomedicine, Far Eastern Federal University, 8 Sukhanova Street, Vladivostok 690950, Russia. [5] Department of Systems Biology, The University of Texas MD Anderson Cancer Center, Houston, TX 77030, USA. [6] Present address: Department of Physics, University of Alberta, Edmonton, AB4-181 CCIS, T6G-2E1, Canada. Correspondence and requests for materials should be addressed to G.B. (email: gabor.balazsi@stonybrook.edu)

Almost two decades after the completion of the Human Genome Project[1], understanding how genes control mammalian cells and organisms remains a daunting task[2,3]. A major factor contributing to this challenge is the complexity of gene regulation at various scales, from underlying molecular mechanisms to large-scale regulatory networks[4,5]. Adding to the conundrum is that genetically identical cells can differ drastically due to microenvironmental and stochastic factors[6–9]. Numerous examples over the last two decades indicate that a population of isogenic cells in the same environment can exhibit single-cell-level stochastic fluctuations in gene expression, also known as gene expression noise[6–9]. Gene expression noise can arise from the intrinsic randomness of underlying biochemical reactions or processes extrinsic to the gene[10]. Two main characteristics of gene expression noise are its amplitude and its memory. The amplitude (often measured by the coefficient of variation or CV) defines how far cells deviate from the average. The memory describes the time for which cells remain deviant once they depart from the average[11,12]. These noise characteristics of a gene depend strongly on the regulatory network that embeds it. Positive regulatory feedback typically increases both the amplitude and memory of noise, while negative feedback tends to have the opposite effect[13], implying that network structure and noise characteristics are deeply intertwined and difficult to separate.

Traditional measurements have generated numerous insights by focusing on the gene expression mean and its cellular effects, but we still need to understand the phenotypic roles of gene expression noise in many circumstances[7,8,14–16]. Likewise, approaches that perturb cells in bulk by over-expression, down-regulation, or knockout try to control only the gene expression mean, without precisely adjusting cell-to-cell stochasticity or considering its phenotypic effects[17,18]. Hypotheses from over a decade ago propose that non-genetic heterogeneity aids cell survival during drug treatment[12,15,19,20] and other forms of environmental stress[21,22]. These effects depend on the amplitude and memory of noise, both of which are network-dependent. The network conferring noise can evolve[23] and noise improves the adaptive impact of beneficial mutations under stress[24]. Studies in human cells seemingly suggest that cellular heterogeneity and gene expression noise in general promote chemotherapy resistance[20,25], evasion of apoptosis[26], and metastasis[27,28]. However, prior demonstration that noise can also be harmful in low stress[19,22] cautions against the generality of these conclusions. Moreover, prior work implies that examining the phenotypic effects of noise requires proper, mean-decoupled noise control[19,29,30], which has not been established for mammalian cells. Therefore, despite the growing interest in the role of mammalian gene expression noise, its precise role in mammalian cell survival and evolution remain open questions. Addressing these questions requires establishing mammalian cell lines that are as similar as possible, differing only in the networks controlling their gene expression noise. To achieve this, one might manipulate the expression of genes by selecting and mixing cells[28], controlling transcriptional regulators, or applying noise-altering chemicals[31]. However, the regulatory networks that control mammalian gene expression are large, complex[32,33] and incompletely known, making predictable and mean-decoupled noise control for specific individual genes in their native context difficult. Thus, unraveling how gene expression noise of specific networks affects mammalian cell evolution remains a serious challenge.

The field of synthetic biology builds bottom-up synthetic regulatory circuits, which often mimic natural network structures[34,35]. While gene expression noise is difficult to control endogenously, simple synthetic gene circuits have been specifically engineered to modulate noise independently of mean gene expression levels in yeast[19,36] and bacteria[29,30]. In such cases, two non-overlapping noise vs mean curves have decoupled noise regimes (Fig. 1a), which consist of decoupled noise points (DNPs) where two different noise values correspond to the same mean. Low-noise gene circuits for this purpose could include synthetic microRNA-based feedforward loops[37–40] or negative autoregulation[41–43]. In contrast, synthetic gene circuits that incorporate positive auto-regulation or ultrasensitivity have high gene expression noise in yeast[44,45] and mammalian cells[46,47]. Enforcing similar means, but different noise levels in yeast indicated (Fig. 1b), consistently with computational models (Supplementary Fig. 1), that noise aids survival in high stress whereas it hinders survival in low stress if the kill curve is sharp. For gradual kill curves, cells with high noise always have a survival advantage regardless of the stress level. Testing the role of network structure and noise in mammalian cell evolution requires a similar control feat. However, genes integrate randomly into mammalian genomes, which can impose locus-dependent effects on gene expression[9,48], compromising rigorous noise control in mammalian cells. Therefore, noise-decoupling gene circuits should be reliably integrated at the same transcriptionally active locus to minimize such locus-dependent effects.

Here, we integrate mammalian-optimized high-noise positive-feedback (mPF) and low-noise negative feedback (mNF) synthetic gene circuits (Fig. 1a) into separate, but isogenic Chinese Hamster Ovary (CHO) cells at the same well-expressed genomic locus by utilizing the Flp-In™ system[49]. By comparing gene expression in CHO cell lines carrying each gene circuit, we establish decoupled noise points with different gene expression noise levels but with similar mean expression. By using these gene circuits to control the expression of the *Puromycin N-acetyl-transferase* (*PuroR* or *pac*) gene that confers resistance to the antibiotic Puromycin, we investigate how mNF and mPF gene expression noise influences mammalian drug resistance evolution. We find that the mPF gene circuit with high *PuroR* expression noise can aid long-term evolutionary adaptation of mammalian cells at the highest stress (Puromycin) level, whereas it has the opposite effect at low stress. Moreover, by withdrawing and re-adding the drug we find that the gene circuit can mutate to adapt stably in mNF cells. On the contrary, cells with the mPF gene circuit do not adapt by intra-network mutations and their resistance is unstable without circuit induction. Overall, combining mammalian synthetic biology with experimental evolution indicates that the noisy mPF network aids adaptation of mammalian cells to high drug levels, while the opposite is true at low drug levels. These findings may have implications for cancer treatment with known regulatory mechanisms of resistance.

## Results

**Developing a high-noise puromycin resistance gene circuit.** To obtain high gene expression noise amplitude and memory, we designed and assembled a Flp-In-compatible version of the positive-feedback (PF) synthetic gene circuit[45]. We integrated this mammalian PF-PuroR (mPF-PuroR or mPF) gene circuit into the well-expressed genomic FRT site of clonal Chinese Hamster Ovary (CHO) Flp-In™ cells to avoid genomic locus-dependent variation in silencing. In mPF-PuroR, the reverse tetracycline Trans-Activator (*rtTA*)[17] binds to Doxycycline (Dox) and activates the transcription of a tricistronic construct consisting of that same *rtTA* regulator, the fluorescent reporter *EGFP*, and the drug resistance gene *PuroR* (Fig. 2a). Thus, with Doxycycline induction, the positive auto-regulatory network increases fluctuations in gene expression within a population of cells. We joined these coding sequences transcriptionally using the self-cleaving Porcine

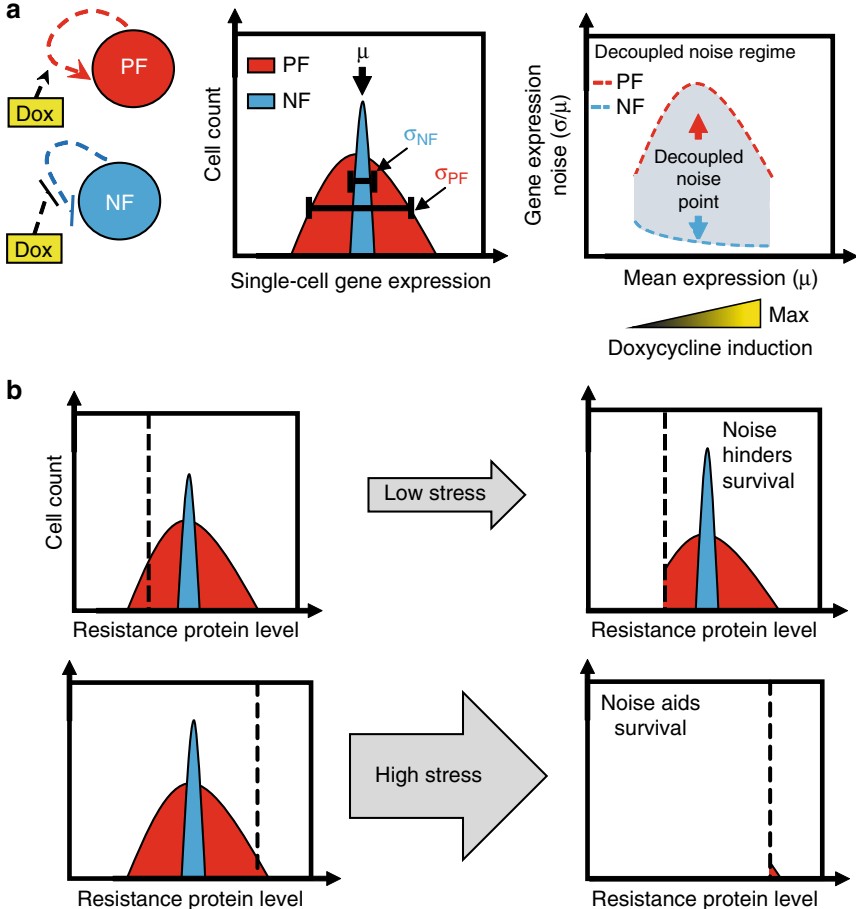

**Fig. 1** Stress-dependent effect of network noise on drug resistance. **a** Tuning the induction (yellow gradient) of mammalian positive (mPF, left, red circle) or negative (mNF, left, blue circle) feedback synthetic gene circuits can confer high and low gene expression noise while the mean expression is identical. This enables decoupling gene expression noise amplitude (middle, standard deviation divided by the mean; $\sigma/\mu$) from the mean within a decoupled noise regime (right, red and blue dashed lines) composed of decoupled noise points (right, red and blue arrows). **b** Schematic depictions to illustrate fractional viability under low or high levels of drug (stress, grey arrow) for cells with high (red distribution) or low (blue distribution) gene expression noise of a drug resistance gene. Relative survival of cells upon drug treatment will depend on network noise relative to the fitness function (dashed black line). If the fitness function is steep, noise hinders survival under low levels of drug while it is beneficial under high levels of drug (Supplementary Fig. 1)

teschovirus-1 2A (P2A) and Thosea asigna virus 2A (T2A) peptides to prevent potential unwanted functional effects from protein fusion[50]. Once translated, the P2A and T2A peptide motifs cleave themselves, leading to the expression of three separated proteins from one transcript. This simple design, with a single common promoter, minimizes the number of genetic components in the mPF-PuroR gene circuit, facilitating genomic integration.

To characterize the expression of the mPF-PuroR gene circuit, we collected single-cell-level *EGFP* fluorescence data at varying Doxycycline levels by flow cytometry. To minimize technical variation from flow cytometry measurements, we normalized this data by correcting for auto-fluorescence and then dividing by the mean of the highest-fluorescence peak from flow cytometry calibration beads (see Data Analysis and Statistics in the Methods). We characterized these normalized *EGFP* fluorescence distributions in terms of their gene expression mean and noise amplitude, quantified by the CV. The mean mPF-PuroR expression dose-response was sigmoidal with a steep response region (Fig. 2b; Supplementary Fig. 2a, c), similar to yeast[45]. Gene expression noise amplitude for uninduced mPF-PuroR cells was low, but then increased markedly upon Doxycycline induction (Fig. 2c; Supplementary Fig. 2b, d). The highest noise amplitude values corresponded to broad, yet visibly unimodal single-cell

expression distributions (Fig. 2d; Supplementary Fig. 3a) in contrast to the bimodal distributions in yeast[45]. The removal of *PuroR* did not impact the performance (noise amplification) of the mPF circuit (Supplementary Fig. 4). To summarize, transferring the mPF-PuroR gene circuit into CHO Flp-In cells led to high noise amplitude with broad, visibly unimodal distributions.

**Low- and high-noise gene circuits decouple noise jointly.** To generate a low-noise gene circuit in the same genomic locus, we also integrated a Flp-In-compatible mammalian negative feedback (mNF-PuroR or otherwise called mNF) gene circuit in the same ancestral CHO cell line (Fig. 3a). With negative feedback, gene expression fluctuations are suppressed[41–43]. We preserved previous optimizations that enhanced gene expression[51]. Again, we joined the humanized Tetracycline repressor (hTetR), the *EGFP* reporter, and *PuroR* genes with P2A and T2A peptide motifs, allowing co-translational separation of the three proteins.

To determine how the gene expression mean and noise amplitude of the mNF-PuroR circuit depend on Doxycycline, we obtained gene expression distributions by flow cytometry. As expected[51], the mNF-PuroR gene expression mean increased linearly with Doxycycline concentrations prior to saturation (Fig. 3b; Supplementary Figs. 2a, e; 5a, b). We observed low gene

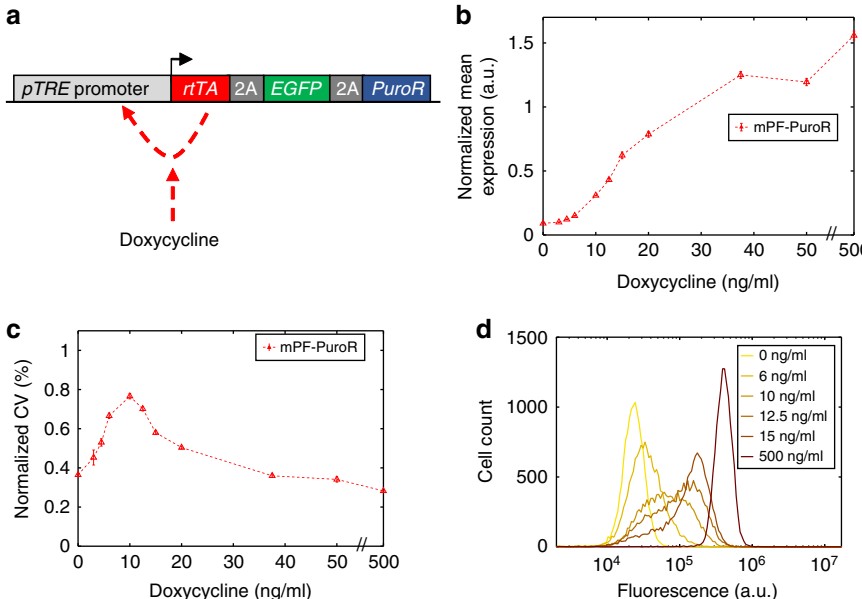

**Fig. 2** Dose-response of the mPF-PuroR gene circuit. **a** Network schematic of the mPF-PuroR gene circuit induced by Doxycycline (Dox), which expresses the reverse tetracycline transactivator (rtTA) regulator, the Puromycin resistance gene (PuroR) and EGFP separated by the self-cleaving 2A elements. The rtTA regulator activates its own expression upon binding Dox (red dashed line). **b** Normalized mean expression under varying levels of Doxycycline induction. **c** Gene expression noise amplitude (normalized coefficient of variation, CV) in response to Doxycycline induction. Error bars denote the standard error of the mean. There is an x-axis break (//) between 50 and 500 ng/mL Doxycycline. All samples were measured in triplicate ($n = 3$). **d** Single-cell gene expression distributions of mPF-PuroR cells with broad peaks at intermediate levels of Doxycycline. The legend displays Doxycycline concentrations per distribution. Distributions are from representative replicates. Source data are provided as a Source Data file

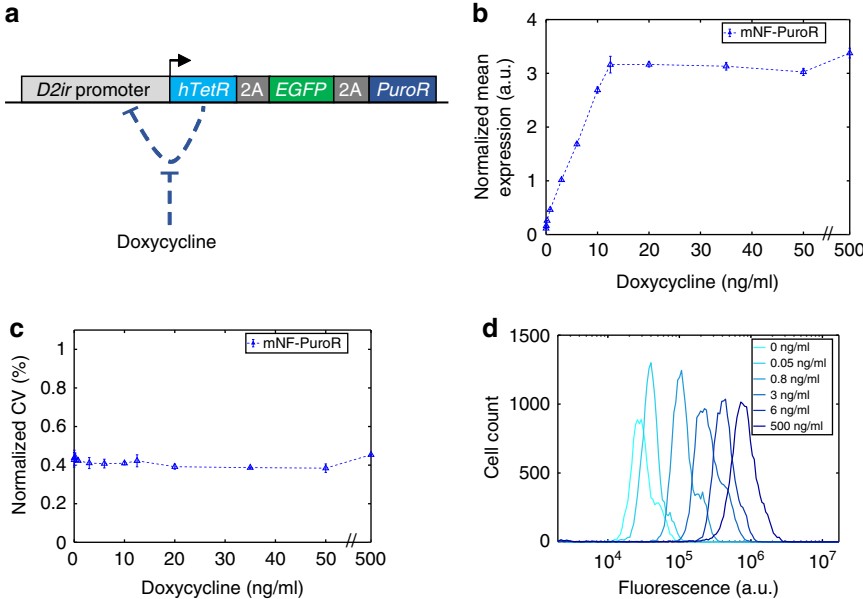

**Fig. 3** Dose-response of the mNF-PuroR gene circuit. **a** The mNF-PuroR gene circuit controls the expression of a Puromycin resistance gene and the EGFP reporter gene through inducible negative auto-regulation (blue dashed line) of a humanized tetracycline repressor (hTetR) gene. The 2A peptides self-cleave after translation. **b** Normalized mean expression of mNF-PuroR cells under varying levels of Doxycycline (Dox). **c** Gene expression noise of mNF-PuroR cells in response to Doxycycline. Error bars denote the standard error of the mean. There is an x-axis break (//) between 50 and 500 ng/mL Dox. All samples were measured in triplicate ($n = 3$). **d** Single-cell gene expression distributions of the mNF-PuroR circuit. The legend indicates Doxycycline concentrations for each distribution. Distributions are from representative replicates. Source data are provided as a Source Data file

expression noise amplitude in response to Doxycycline (Fig. 3c; Supplementary Fig. 2b, f), in agreement with narrow gene expression distributions (Fig. 3d; Supplementary Fig. 3b). Removing *PuroR* did not affect the performance of the mNF gene circuit (Supplementary Figs. 5c, d; 6).

To test whether noise-mean decoupling is possible with mNF-PuroR and mPF-PuroR, we sought Doxycycline induction levels where the mean expression of the two gene circuits were similar while the differences in noise amplitude were high (Fig. 1a). To identify such a decoupled noise regime from the dose-response,

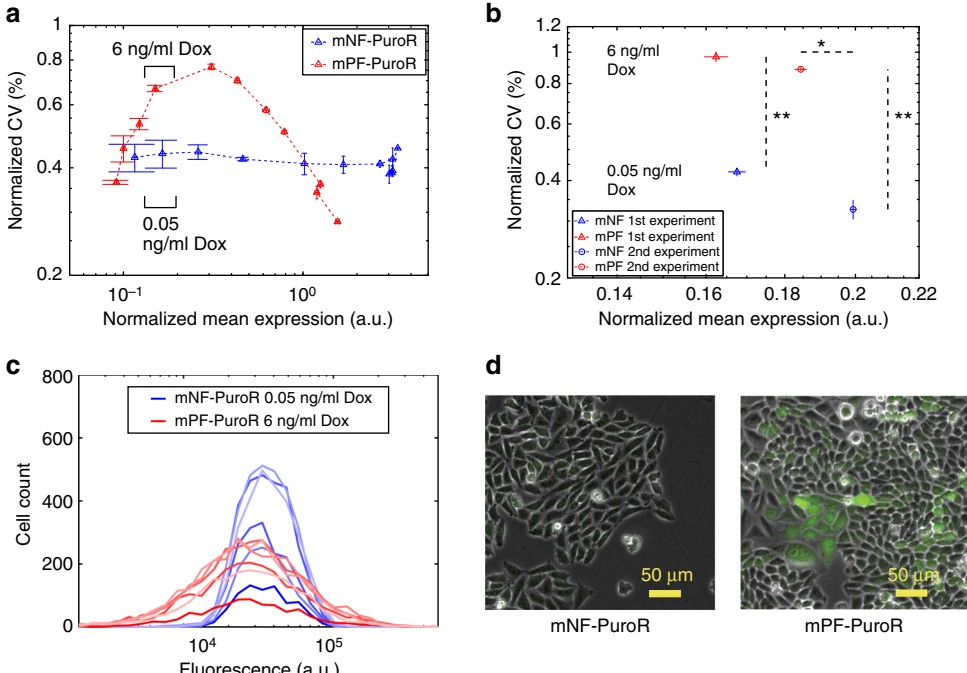

**Fig. 4** Decoupled noise regime and decoupled noise points before treatment. **a** Plotting the noise (coefficient of variation, CV) as a function of normalized mean gene expression for both gene circuits revealed two decoupled noise regimes. Black brackets indicate the expression range for the Doxycycline (Dox) concentrations used for noise-mean decoupling in (**b**). Error bars represent the standard error of the mean ($n = 3$). **b** Decoupled noise points (DNPs) at the beginning of the two drug treatment experiment sets. The noise was significantly different between gene circuits for both sets (**$p$ value $= 0.0022$, $n = 6$, two-tailed Mann–Whitney $U$ test). The mean expression was not significantly different for set 1 (~3%; $p$ value $= 0.0931$, $n = 6$, two-tailed Mann–Whitney U test) while it had significance for set 2 (~8%; *$p$ value $= 0.0022$, $n = 6$, two-tailed Mann–Whitney $U$ test). The dashed lines display the range of statistically significant differences for gene expression noise and mean expression between the two gene circuits. **c** Gene expression distributions at the DNP from the first experiment set, filtered as described in the Methods. **d** Images of cells at the decoupled noise point from the first experiment set. The two-tailed Mann–Whitney U test inferred significance at $p$ values < 0.05. Source data are provided as a Source Data file

we analyzed noise amplitude as a function of mean expression (Fig. 4a). We observed two decoupled noise regimes, one before and one after the mNF-PuroR and mPF-PuroR noise-mean curves intersect at high mean expression. Each regime has a set of decoupled noise points (DNPs) where circuits can have matching mean expression while ensuring distinct noise amplitudes. Although we only measure *EGFP* expression directly, these measurements should reflect *PuroR* expression, including noise decoupling because the two proteins are co-translated (Supplementary Fig. 7c, d, e). Besides the noise amplitude, we also studied the memory, estimating the rate at which cells moved within the distributions. Flow-sorting high- or low-expressing subpopulations for both gene circuits at DNP induction levels indicated that cells with the high-noise mPF-PuroR circuit have higher memory (~2 days) than cells with the low-noise mNF-PuroR circuit (~1/2 day) (Supplementary Fig. 7; Supplementary Table 1). Overall, both cell lines were equivalent except for the noise-controlling constructs, each integrated as a single copy (Supplementary Fig. 8) into the same genomic locus of a clonal cell line. Thus, the decoupled noise regimes provide DNPs to jointly control *PuroR* gene expression noise amplitude and frequency and test their role in drug resistance evolution.

**The noisy network can aid or hinder drug resistance evolution.** To uncover the role of noise-controlling mNF and mPF networks in mammalian drug resistance evolution, we decoupled noise from mean *PuroR* expression in isogenic CHO cells. By following these cells in constant inducer concentrations through parallel flow cytometry and microscopy (Fig. 5a), we identified a DNP for mNF-PuroR and mPF-PuroR at 0.05 and 6 ng/mL Doxycycline,

respectively, in two experimental sets. At the DNPs, the means differed by less than 10%. The gene expression noise amplitudes were significantly distinct (**$p$ value $= 0.0022$, $n = 6$, two-tailed Mann–Whitney $U$ test) prior to treatment in both sets (Fig. 4b; Supplementary Fig. 9a, b). Accordingly, the high-noise mPF-PuroR expression distribution consisted of a wide, visibly unimodal peak while the low-noise mNF-PuroR peak was narrow (Fig. 4c; Supplementary Fig. 9c, d), which is apparent in imaging (Fig. 4d). Since the mNF-PuroR mean exceeded slightly the mPF-PuroR mean (non-significantly in the first experiment and significantly, but still within 10%, in the second experiment; $p$ value $= 0.0931$ and *$p$ value $= 0.0022$, respectively, $n = 6$, two-tailed Mann–Whitney $U$ test), observing better mPF-PuroR survival should strengthen the evidence for noise-aided drug tolerance and subsequent resistance.

After preparing six mNF-PuroR and mPF-PuroR replicates at the DNP, we introduced various concentrations of Puromycin and performed sets of evolution experiments, each lasting until the adapting replicates have reached confluency. In the first experiment set, we maintained Puromycin concentrations of 0, 10, and 22.5 μg/mL while in the second set, we kept cells in Puromycin concentrations of 35 and 50 μg/mL. A total of five mPF-PuroR replicates (3 under 35 μg/mL Puromycin and 2 under 50 μg/mL Puromycin) were lost during sample maintenance.

To study the adaptation of CHO cells with low- and high-noise networks to Puromycin treatment, we constructed population-level adaptation curves at 0, 10, 22.5, 35, and 50 μg/mL Puromycin by detecting and counting single cells from daily microscope images. After examining these adaptation curves, we observed immediate growth without adaptation for low

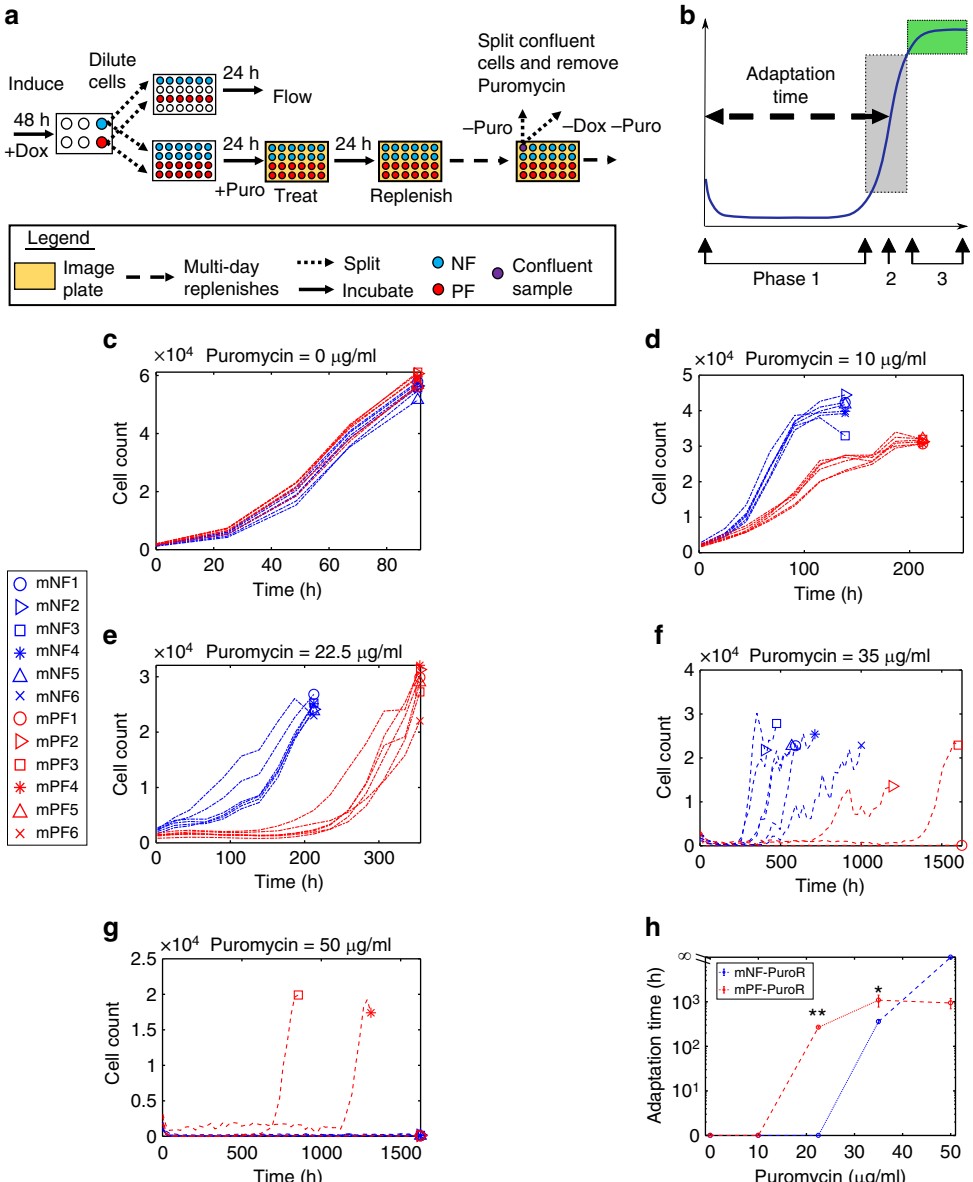

**Fig. 5** Noise hinders resistance under low stress but aids it under high stress. **a** Experimental workflow of the Puromycin (Puro) treatment assay. Cells induced with Doxycycline (Dox) were treated with Puromycin in a parallel series of plates for imaging or for flow cytometry. Upon confluency, Puromycin was removed temporarily (see Fig. 7). **b** Illustration of a representative growth curve with three growth phases: (1) growth suppression, (2) regrowth (gray box), and (3) saturation (green box). **c–g** Growth curves for cells initially tuned to the DNPs under 0 (**c**), 10 (**d**), 22.5 (**e**), 35 (**f**), and 50 (**g**) μg/mL Puromycin. Dash-dot growth curves indicate data from the first experimental set while dash-dash growth curves are from the second experiment set. **h** Mean adaptation times corresponding to (**c–g**) (**\*\***p value = 0.0022, n = 6, two-sided Mann–Whitney U test; \*p value = 0.0238, n = 3 for mPF-PuroR and n = 6 for mNF-PuroR, two-sided Mann–Whitney U test). The statistical test on the data from 35 μg/mL Puromycin included the non-growing mPF replicate 1 (infinite replicate adaptation time), which is not included in the mean (n_mean = 2 for mPF-PuroR) in (**h**). The adaptation time error bars represent the standard error of the mean with replicates as described above. The two-tailed Mann–Whitney U test inferred significant differences at p values < 0.05. Source data are provided as a Source Data file

Puromycin doses, while for moderate to high Puromycin doses, the curves were initially flat, and fast growth resumed with a delay that increased with stress severity. We defined as adaptation only this latter behavior, which revealed three different phases of the adaptation process (Fig. 5b): (1) growth suppression while the curve stayed mostly flat; (2) fast regrowth when the curve rose (grey region); and (3) saturation when the curve flattened again at confluency (green region). Based on these phases, we analyzed the adaptation time, which we defined as the time required for initially suppressed cells to reach half-saturation (indicated by a dashed arrow in Fig. 5b). Interestingly, the

length of the growth suppression phase became more variable between replicates of each circuit upon increasing Puromycin concentrations (Fig. 5c–g).

We first investigated how decoupled *PuroR* expression noise and the corresponding networks affect the adaptation time at various levels of Puromycin. We calculated adaptation times only for replicates whose moving average net growth rates (estimated for every 3 timepoints on the adaptation curves) fell to 0 or below at least once (Supplementary Fig. 10). Based on this definition, CHO cells with both gene circuits grew without adaptation at 0 and 10 μg/mL Puromycin (Fig. 5h; Supplementary Fig. 10a, b).

Under 22.5 µg/mL Puromycin, the high-noise mPF-PuroR cells adapted after a delay (**$p$ value = 0.0022, $n = 6$, two-tailed Mann–Whitney U test) while the low-noise mNF-PuroR cells grew without adapting (Fig. 5h; Supplementary Fig. 11a). Likewise, the low-noise mNF-PuroR replicates adapted faster (*$p$ value = 0.0238, $n = 6$ for mNF-PuroR and $n = 3$ for mPF-PuroR, two-tailed Mann–Whitney U test) with a shorter delay than the mPF-PuroR replicates (2 out of 3 surviving) under 35 µg/mL Puromycin (Fig. 5h; Supplementary Fig. 11b). Interestingly, cells with the high-noise network tended to exhibit morphological diversity, including signs of polyploidy (Supplementary Figs. 12–13) during adaptation. In contrast to the lower drug concentrations, at 50 µg/mL Puromycin all mNF-PuroR replicates perished whereas half (two) of the mPF-PuroR replicates eventually adapted and recovered (Fig. 5h; Supplementary Figs. 11c; 14–15). Importantly, visual inspection indicated that mNF-PuroR cells completely vanished from the culture wells despite having slightly higher pre-treatment mean *PuroR* expression (Supplementary Fig. 14), indicating their inability to adapt to the highest Puromycin concentration.

Overall, the adaptation times indicate that the noisy mPF network promotes evolutionary adaptation compared to mNF at high stress, while the reverse is true for low stress, which is consistent with the effects of noise on survival immediately after treatment for steep kill curves (Supplementary Fig. 1). Therefore, the noisy mPF network affects long-term mammalian drug resistance evolution similarly to noise-dependent short-term survival of other cell types[19]. The most pronounced evolutionary benefit from the noisy mPF network occurs at the highest stress level, but it is not directly related to its preexisting gene expression fluctuations[12], as the adaptation time to regrowth (weeks) greatly exceeds the memory of preexisting *PuroR* expression fluctuations (~2 days).

**Computational modeling infers mechanisms for drug resistance.** To investigate whether gene expression noise differences alone explain the experimentally observed adaptation to drug treatment, we used a stochastic gene expression model that decouples noise magnitude and fluctuation relaxation time from the mean[12]. To model cell population size over time, we performed stochastic simulations of growing cells with gene expression based on the Ornstein-Uhlenbeck process[52], fixing mean gene expression levels, but varying gene expression noise and relaxation times to match the two gene circuits. These simulations indicated that the preexisting noise properties of the gene circuits alone without additional cellular states could not capture the long experimental adaptation times followed by fast regrowth for prolonged, high Puromycin drug treatment (Supplementary Fig. 16). If slow gene expression fluctuations would underlie drug resistance without any phenotypic switching to other cell states, the computational model assumingly suggests slow but visible growth in a few days, as opposed to the experimentally observed multi-week delays without growth.

Next, to better capture experimentally observed long-term evolutionary adaptation behaviors at high stress, we developed a more complex stochastic population/evolutionary dynamics model[53]. This model assumed additional cellular states based on short-term experimental data before and immediately after drug treatment. Specifically, we assumed that cells die if their *PuroR* concentration is below a specific Puromycin-dependent threshold, which we estimated from the initial fraction of cells surviving Puromycin treatment [Eq. (3)–(5), Methods]. We partitioned the remaining surviving cells into stress-induced persister cells[54,55] that neither grow nor die and preexisting, nongenetically drug-resistant cells that grow in the presence of Puromycin [Eq. (6), Methods] (Fig. 6a). Thus, upon initial drug treatment, three different cell types exist within the cell population: dead ($D$), persister ($P$), and nongenetically drug-resistant ($N$) cells. We allowed phenotype switching between $P$ and $N$ cells. Additionally, we assumed that over time $P$ cells and $N$ cells can give rise to a fourth, stable (genetically or epigenetically) drug-resistant ($G$) cell type. Though the growth rates of $N$ and $G$ cells were similar, the death rates of $N$ cells were increasingly greater than $G$ cells for higher Puromycin concentrations (Supplementary Table 2). Gene expression noise and the drug concentration imposing selective pressure determine the $D$, $P$, and $N$ cell population proportions shortly after treatment, and consequently, the ultimate evolutionary outcome in these simulations (Fig. 6b–f). As Puromycin concentrations increase, the number of $D$ cells increases accordingly at the expense of surviving cells. Among surviving cells, the frequency of $P$ cells increases with stress levels until all surviving cells are $P$ cells at very high stress (Supplementary Fig. 17). $G$ cells emerge subsequently at rates reflecting the numbers of their $P$ and $N$ cell precursors.

To investigate various adaptation scenarios, we scanned cell-type switching and mutation rate parameters ($r_{N,P}$, $r_{P,N}$, $r_{G,N}$, $r_{G,P}$; Supplementary Table 2) over four orders of magnitude in the population dynamics model. Interestingly, most of the scenarios could not capture the adaptation times in all Puromycin concentrations. For instance, in models that excluded $P$ to $G$ conversions, the mPF cell population grew immediately (without adaptation), although slowly in 22.5 µg/ml Puromycin (Supplementary Fig. 18), in disagreement with experimental results (Fig. 5e, h). Models that allowed $P$ to $G$ conversion captured the experimental adaptation time dynamics in all tested Puromycin conditions (Fig. 6g). Therefore, the experiments and models jointly support that resistance to high Puromycin levels occurs by drug-induced formation of persister-like cells serving as reservoirs for fast-growing, heritably drug-resistant mutants. Mutant antibiotic tolerant non- or slow-growing cells have previously been shown to precede genetic drug resistance during intermittent antibiotic exposure in bacteria[56]. On the other hand, at lower Puromycin levels adaptation may arise from preexisting nongenetic phenotypic variability, as nongenetically drug-resistant cells survive and grow immediately upon drug treatment.

**Drug removal and sequencing suggest how mNF evolves.** As the computational model indicated, after initial cell death, adaptation to Puromycin stress could occur by multiple different mechanisms depending on the stress level. Specifically, at low stress, nongenetically resistant cells could continue growing, and eventually reestablish the population without any mutations or other heritable alterations[12]. Alternatively, at high stress, cells can acquire heritable (genetic or epigenetic) drug resistance alterations after a significant delay, leading to stable resistance. Heritable mechanisms could be endogenous (based on native mechanisms independent of *PuroR* gene expression), or *PuroR*-dependent, elevating *PuroR* expression to a level sufficient for resistance. However, for all mNF and mPF replicates evolved at the highest 3 stress conditions, induced *PuroR* expression increased and stayed far above pre-treatment levels. Therefore, we concluded that adaptation always relied on elevated *PuroR* expression.

*PuroR*-dependent mechanisms could occur inside or outside the synthetic gene circuit and may depend on network induction. To distinguish between such possibilities (Supplementary Fig. 19), and to formulate hypotheses about the nature of molecular events contributing to evolutionary adaptation, we removed Puromycin temporarily and then re-added it again for cells that have adapted under 22.5, 35, and 50 µg/mL Puromycin.

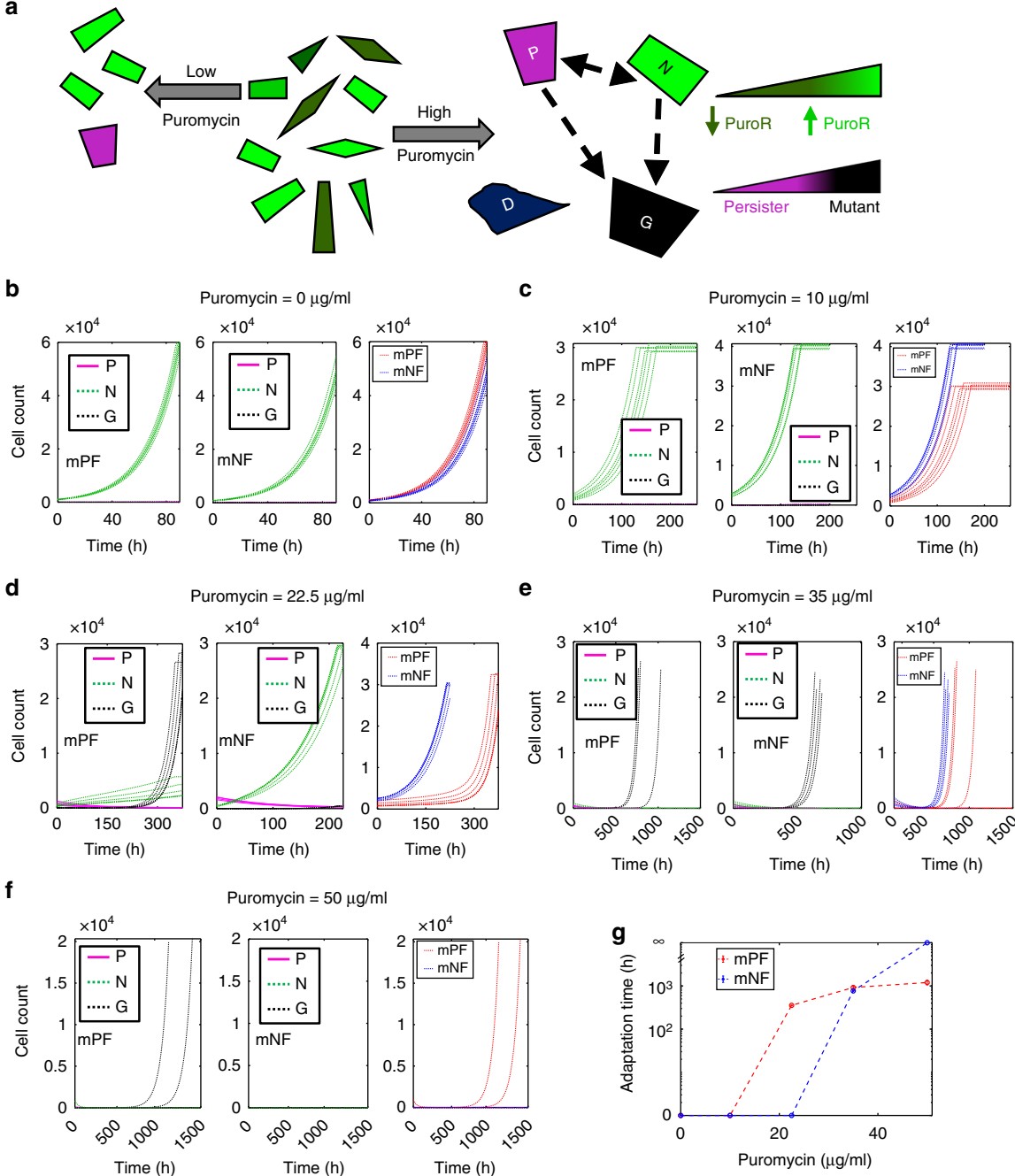

**Fig. 6** Modeling adaptation of mPF and mNF cells in various Puromycin concentrations. **a** Schematic depicting the effects of Puromycin concentration on CHO cell population composition and survival. Nongenetically Puromycin-resistant cells (green cells - brighter cells have higher PuroR expression level and are therefore more resistant) and nongrowing persister cells (magenta cells) can switch phenotypes (dashed bidirectional arrow). Persister cells and growing nongenetically resistant cells can also become stably Puromycin-resistant cells (black cells). When no Puromycin is present, a clonal population with heterogeneous gene expression exists (center). Under low Puromycin treatment conditions (left arrow), cells with low PuroR expression perish and a small fraction of the surviving clonal cells become persister cells. For high Puromycin treatment conditions (right arrow), only cells with high PuroR expression levels can survive drug treatment while the rest die (dark blue cells), and a higher fraction of the surviving cells become persisters. As persister and nongenetically resistant cells can become stably drug-resistant, the population on the right panel becomes increasingly heterogeneous over the course of treatment. **b**–**f** Representative growth curves for simulated mPF-PuroR and mNF-PuroR CHO cell populations under (**b**) 0, (**c**) 10, (**d**) 22.5, (**e**) 35, and (**f**) 50 μg/mL of Puromycin. Growth curves shown in panels in (**b**–**f**) correspond to: mPF subpopulations (left), mNF subpopulations (center), and full mPF and mNF populations (right). **g** Adaptation times corresponding to the mPF-PuroR and mNF-PuroR populations shown in panels (**b**–**f**). The model is described in the "Methods" section and parameter values are given in Supplementary Table 2

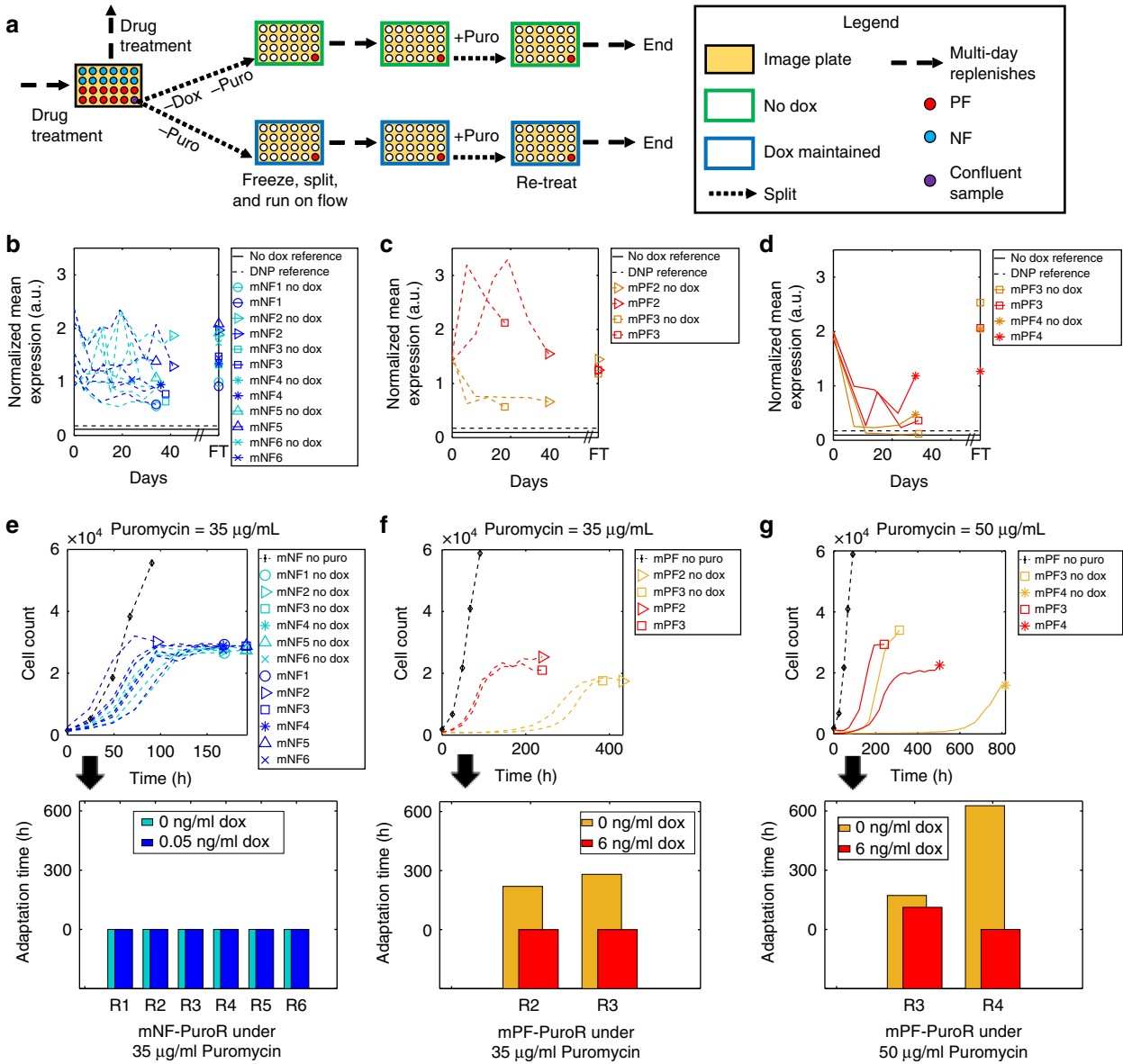

**Fig. 7** Temporary drug removal suggests PuroR-dependent resistance mechanisms. **a** Schematic for the drug removal and retreatment experiment. Doxycycline (Dox) was removed or maintained simultaneously with drug removal. **b** Mean expression for mNF-PuroR during temporary removal of 35 μg/mL Puromycin and after final retreatment (FT). **c**, **d** Mean expression for mPF-PuroR after removal of (**c**) 35 and (**d**) 50 μg/mL Puromycin and after final retreatment (FT). **e**–**g** Growth curves (top) and adaptation times (bottom) during re-treatment for (**e**) mNF-PuroR cells under 35 μg/mL, and mPF-PuroR cells under (**f**) 35 and (**g**) 50 μg/mL Puromycin. The solid and dashed black lines in panels **b**–**d** indicate uninduced and induced baseline mean expression levels, respectively, prior to treatment with Puromycin. The black growth curves are averaged untreated cell counts from Fig. 5c. Source data are provided as a Source Data file

Moreover, to test whether gene circuit induction was necessary for resistance, we split each evolved replicate into two separate wells, culturing them either without Doxycycline (uninduced) or with Doxycycline (induced) at the same concentration as before Puromycin removal (Fig. 7a).

Next, we studied the behavior of mNF-PuroR replicates after removal of 35 μg/mL Puromycin. All uninduced and induced mNF-PuroR replicates maintained constant *PuroR* expression levels well above corresponding induced but untreated ancestral cells for ~ a month (Fig. 7b; Supplementary Figs. 20a; 21–22), suggesting that inducer-independent, high *PuroR* expression has evolved. Accordingly, all uninduced and induced mNF-PuroR replicates grew without adaptation upon Puromycin re-addition, much quicker than their Puromycin-treated ancestors, further

supporting stable, induction-independent drug resistance in each population (Fig. 7e).

To examine how inducer-independent, *PuroR*-dependent resistance arose in the mNF-PuroR circuit, we sequenced the gene circuit from the six induced replicates after drug removal at 35 μg/mL Puromycin. In replicate 2, we found an indel in *hTetR* that can reduce binding affinity to *tetO2* sites by 1000-fold[57] (Supplementary Fig. 23). Therefore, this mutation likely compromised repressor functionality, leading to high *PuroR* expression and drug resistance. In replicate 3, we found a single nucleotide polymorphism in the distal region of the promoter (Supplementary Fig. 24). Furthermore, the CRISP-ID[58] genotyping algorithm uncovered in replicate 1 two mutant variants in the same distal promoter region as in replicate 3 (Supplementary

Fig. 25). Both variants contain the same mutation as replicate 3, suggesting that both arose by selection for elevated *PuroR* expression. Therefore, mutations abrogating *hTetR* repression seem to occur repeatedly, possibly because random mutations are typically functionally deleterious rather than beneficial for proteins[23]. Here, mutations deleterious for hTetR protein function are beneficial for cellular drug resistance. Despite identical phenotypes (stable inducer-independent expression), we found no mutations in the mNF-PuroR circuit from replicates 4–6 (Supplementary Fig. 26). In summary, the mNF gene circuit adapts through intra-circuit mutations or extra-circuit heritable alterations that corrupt *hTetR* repressor function to confer elevated, inducer-independent *PuroR* expression (Supplementary Fig. 27).

Finally, we followed the same procedure to gain insights for evolution in 22.5 μg/mL Puromycin, the lowest stress level, where mNF-PuroR cells grew instantaneously. At this stress level, after drug removal both uninduced and induced mNF-PuroR mean expression levels reverted quickly towards their pre-treatment levels (Supplementary Fig. 28a). This indicated non-genetic drug resistance purely from non-uniform *PuroR* expression. Sequencing revealed no intra-circuit mutations, further supporting these conclusions (Supplementary Fig. 29a). Overall, the lack of intra-circuit mutations and the quick reversion to pre-treatment expression levels at 22.5 μg/mL Puromycin suggest nongenetic drug resistance mechanisms relying on preexisting Doxycycline-dependent *PuroR* expression variability, as predicted computationally at sufficiently low stress levels (Fig. 6d; Supplementary Fig. 16).

**Sequencing reveals mPF adaptation without circuit mutation.**
To investigate molecular adaptation mechanisms of mPF cells to 35 μg/mL Puromycin, as for the mNF gene circuit, we sequenced the high-noise mPF-PuroR gene circuit, but found no mutations (Supplementary Figs. 30–31) for any replicate. Therefore, extra-circuit heritable alterations must confer resistance by *rtTA* induction-dependent or independent mechanisms. To distinguish between these possibilities, as before, we compared induced versus uninduced cell count and gene expression time courses for mPF-PuroR replicates during drug removal and re-addition. In contrast to cells with the mNF-PuroR circuit, uninduced mPF-PuroR replicates showed signs of regulator (*rtTA*) induction-dependent adaptation, as their expression dropped closer, albeit not completely down to ancestral levels (Fig. 7c; Supplementary Figs. 20b; 32a, b), and they failed to grow initially after drug re-addition, adapting with a long delay (Fig. 7f). Induced mPF-PuroR cells maintained their expression well above induced and uninduced ancestral cells (Fig. 7c; Supplementary Figs. 20b; 32c, d) and regrew quickly without adaptation upon retreatment (Fig. 7f). Together with the lack of intra-circuit mutations and reacquisition of drug sensitivity after Doxycycline removal, the evidence supports *rtTA* induction-dependent extra-circuit alterations that elevate *PuroR* expression to resist 35 μg/mL Puromycin (Supplementary Fig. 33).

At the highest level of 50 μg/mL Puromycin, two mPF-PuroR replicates recovered, demonstrating the evolutionary benefit of the noisy mPF-PuroR gene circuit over mNF-PuroR at very high stress levels. Once again, sequencing did not reveal any intra-circuit mutations (Supplementary Fig. 31). The expression of uninduced mPF-PuroR replicates dropped closer to the baseline DNP mean over ~10 days (Fig. 7d; Supplementary Figs. 20c; 32e, f). Interestingly, for uninduced replicate 3 expression dropped farther down, and re-adaptation to Puromycin occurred even in the induced condition (Fig. 7g). Moreover, uninduced replicate 4 cells required more time to adapt upon retreatment compared to replicate 3, despite slightly higher expression levels, which

suggests distinct heritable alterations contributed to resistance in each replicate. Overall, we found evidence of distinct extra-circuit heritable inducer-dependent mechanisms maintaining high *PuroR* expression at 50 μg/mL Puromycin (Supplementary Fig. 33).

Finally, we applied similar criteria to gain insights for 22.5 μg/mL Puromycin, the lowest stress level where mPF-PuroR replicates adapted with a moderate delay. All induced mPF-PuroR replicates maintained their *PuroR* expression above the ancestral levels (Supplementary Fig. 28b), but uninduced replicates dropped close to baseline, indicating stable *PuroR* expression-dependent mechanisms of resistance requiring *rtTA*-induction. Accordingly, uninduced mPF-PuroR replicates failed to grow initially during retreatment, showing signs of adaptation (Supplementary Fig. 34b, d), as opposed to induced replicates, which grew instantaneously. Gene circuit sequencing revealed no mutations (Supplementary Fig. 29b), suggesting extra-circuit heritable alterations contributing to adaptation. Overall, in mPF-PuroR we find evidence for extra-circuit, inducer- and *PuroR*-dependent mechanisms of adaptation (Supplementary Fig. 35).

## Discussion

Over a decade ago, hypotheses emerged on gene expression noise contributing to chemotherapy resistance[15]. Noise would ensure initial survival in a memory-dependent manner[12], enabling cancer cells or microbes to then develop genetic resistance. Recent studies have raised further awareness on cellular heterogeneity and gene expression noise, implying a general benefit for cell populations to overcome drug resistance or metastatic barriers[20,25–28]. However, earlier evidence for the harmful effects of noise in low stress[19,22] cautions against generalizing these recent observations. In fact, to rigorously study phenotypic effects of noise requires two cell populations with similar means, but different noise levels[19,29,30], which was lacking for mammalian cells. Without such control, we cannot exclude the possibility that the fitness benefit is from higher mean expression. Therefore, how gene expression noise affects mammalian cell survival and evolution remained open questions, addressing which required isogenic mammalian cells with mean-decoupled noise control. We established such control with high- and low-noise gene circuits to study how network noise contributes to drug resistance evolution in CHO cells. While earlier work in yeast indicated that noise can aid or hinder short-term survival depending on the balance between drug dose and resistance protein levels[19,59], the evolutionary effects of noise are only recently being unraveled[23,24]. Here, by experimentally evolving synthetic gene circuit-harboring CHO cells in Puromycin, we show that noisy mPF networks aid evolution at high stress as previously hypothesized[15], but also hinder evolution at low stress, mimicking the effects of noise on short-term survival[19].

We combined experimental evolution and synthetic gene circuits to drive evolutionary adaptation in mammalian cells. Since the pioneering studies of prokaryotic experimental evolution[60], the field has expanded into yeast[23,24,61] and fruit flies[62]. Mammalian cell evolution studies are timely and relevant to cancer[63], but they are rare and have not involved synthetic gene circuits. Experimental evolution of artificial gene circuits in microbes[23,24] provided mechanistic insights across multiple scales of time and biological organization, by reducing the influence of complex and incompletely known native gene regulatory networks. Thus, synthetic gene circuits facilitate the development of predictive models that reveal unexpected, emergent effects, which would be more difficult to derive for natural gene networks.

The experimental system we developed is a feasible model[63] for the long-term evolutionary response of cancer cells to

translational inhibitors. Puromycin compromises protein synthesis in a broad range of cell types, like emerging cancer therapeutics targeting mRNA translation[64,65]. Moreover, Puromycin itself has been proposed as a potent anticancer agent specifically released from a prodrug in cancer cells[66]. Considering that over 80% (19,711/24,383) of the predicted CHO protein-coding genes have homologs in human[67], studying drug resistance evolution in this cell line should be as relevant as mouse cell line models of drug resistance are to human cancers.

Overall, the data suggest that at the highest stress levels that cause prolonged growth suppression, only cells with high-noise mPF networks recover ultimately through stable, but unknown extra-circuit genetic or epigenetic drug resistance mechanisms. At milder stress levels, cells with the low-noise mNF network adapt partly by mutating the circuit to abrogate repressor function. Surprisingly, adapting CHO cells always take advantage of the non-native *PuroR* gene. The mechanisms vary, and most likely include direct *PuroR* upregulation by intra- or extra-circuit alterations. The intra-circuit mutations or lack thereof reflect the fact that random evolutionary changes can more easily disrupt repression than facilitate activation. The exact extra-circuit heritable mechanisms behind the evolutionary adaptation remain to be studied as whole-genome and -transcriptome sequencing of CHO cells advance[68].

We used different (mNF and mPF) networks to control noise properties, keeping the role of networks and noise intertwined. We think noise properties (amplitude and memory and then switching to a persister state) should be more relevant for initial survival, when the protein level fluctuations make the difference between survival and death. On the other hand, network topology (repression versus activation of drug resistance) and its modes of beneficial alterations seem to matter more at longer, evolutionary time scales. In the future, it will be interesting to try controlling noise while minimizing differences in network topology[36], to separate better the evolutionary effects of networks and gene expression noise.

Comparing the experimental evolution time courses with the evolutionary model and sequencing results suggested that persister cells convert to stably resistant proliferating cells at high stress levels. The mammalian drug-tolerant persister state could derive from a chromatin-mediated transition, which previously has shown sensitivity to HDAC inhibitors[55], or could depend on *GPX4* expression[54]. However, in these experiments persister simply means cells that neither divide, nor die in stress – mediated by many possible mechanisms, such as the formation of polyploid cells[69], as we noticed. Nonetheless, the successful elimination of low-noise populations without resistance at high stress levels provides hope for noise-controlling treatment strategies in cancer, like HIV-infected cells[31].

## Methods

**Plasmid construction**. Plasmids integrated into CHO Flp-In™ cells (Invitrogen, R758-07) were constructed using restriction cloning on commercial and custom vectors and constructs. The mNF and mPF plasmids integrated into the genomic FRT site with the aid of Flp-recombinase expressed from the pOG44 vector (Invitrogen, V600520). The addition of T2A::*PuroR* to both plasmids resulted in mNF-PuroR and mPF-PuroR constructs. The molecular cloning extensively used overlap PCR extension to fuse DNA pieces together. For a detailed description of plasmid construction, see Supplementary Methods and Supplementary Table 3 for cloning primers.

**Cell culture and dose-response**. Chinese hamster ovary (CHO) cells with the single stably integrated FRT site (Invitrogen, R75807) were grown in Ham's F-12 Nutrient Mix (Gibco, 11765) with 10% fetal bovine serum (Gibco, 10437) and 100 U/mL Penicillin and Streptomycin (Gibco, 15140). Hygromycin B (Invitrogen, 10687-010) at 700 µg/mL was used as a selection agent that killed untransfected cells and cells with randomly integrated constructs (see below). Doxycycline (Fisher Scientific, BP26531) stock solution was stored at −20 °C, and diluted in media

at 4 °C storage for no longer than 7 days after initial preparation. Cells were passaged by washing with 1X Dulbecco's phosphate-buffered saline (DPBS) without calcium or magnesium (Life Technologies, 14190250), incubating cells with 0.25% Trypsin-EDTA (Life Technologies, 25200056) up to 5 min in 37 °C with 5% $CO_2$, neutralizing any proteases with media and growth serum, and then cells were transferred to a new flask or tube. Prior to imaging experiments, CHO cell nuclei were stained with the live cell dye NucBlue (Invitrogen, R37605) at a concentration of 1 droplet per 90 mL of media.

**Transfection and flpase-integration**. For the genetic constructs containing the *PuroR* gene, CHO Flp-In cells were transfected with plasmid DNA (up to 5 µg) using the Lipofectamine 3000 reagent (Life Technologies, L3000008) according to the manufacturer protocol. Plasmid DNA for mNF-GFP and mPF-GFP was introduced into cells using the Nucleofector™ 2b device (Lonza, Walkersville, MD), per manufacturer protocol, using 5–10 × 10⁶ cells, plasmid DNA (1–5 µg), and relevant buffers (Solution T, and program V-23). Site-specific integration of synthetic gene circuits was achieved by co-transfecting the pOG44 plasmid expressing Flp-recombinase with the Flp-expression vectors that encode an FRT-tagged Hygromycin B resistance gene without a start codon. Upon selecting with Hygromycin B, the resistance gene acts as a positive-selection promoter trap, which provides the resistance gene with a start codon only upon successful integration at the genomic FRT site, thus leading to survival. The clonal CHO populations were derived from bulk-transfected cells by fluorescence-activated cell sorting (FACS).

**Flow cytometry**. The BD Accuri™ C6 benchtop flow cytometer measured *EGFP* fluorescence data from single CHO cells. Before measuring expression, cell samples were trypsinized, neutralized with media, centrifuged at 300 × *g* for 5 min, resuspended in 1X DPBS, and then strained into a 5 mL polystyrene round-bottom tubed (VWR, 21008948) for subsequent flow cytometry data collection. Up to 20,000 events were gated for analysis. Samples treated with drug typically had lower cell counts. In the case of the flow-sorting experiments, cells were sorted by the FACSAria III instrument at the Stony Brook School of Medicine Research Flow Cytometry Core facility. The BD FACSCalibur Cell Analyzer flow cytometer at the facility measured expression after sorting in memory estimation experiments (see Supplementary Methods).

**Data analysis and statistics**. We developed custom MATLAB scripts to gate and analyze flow cytometry data. Cells were adaptively gated with a density-threshold fit of log-transformed SSC and FSC values per sample to exclude debris (see Supplementary Fig. 36 and Supplementary Methods). Fluorescent events that were less than 2,000 arbitrary units were filtered prior to normalization using the following formula:

$$EGFP^i_{filtered} = EGFP^i_{raw}|(EGFP^i_{raw}>EGFP_{cutoff}), \qquad (1)$$

where $EGFP^i_{filtered}$ is the filtered fluorescence for individual cells (i) that have an unsilenced circuit, $EGFP^i_{raw}$ is the raw fluorescence from any cell (i) with the circuit, and $EGFP_{cutoff}$ is a constant fluorescence threshold (2,000 arbitrary units) above auto-fluorescence but below uninduced, basal circuit expression. The $EGFP_{cutoff}$ was shown to threshold non-fluorescent cells from cells with uninduced circuits based on the threshold value exceeding three standard deviations higher than the auto-fluorescence mean. Additionally, the non-expressing peak and basal expression peak flank a local minimum (Supplementary Fig. 2) that aligns with the $EGFP_{cutoff}$. The filter was justified by the lack of transitions from the small subpopulation of non-expressing cells to the basal expression level for both circuits (Supplementary Fig. 37).

To estimate technical variation in fluorescence under flow cytometry between experiments, we measured fluorescent values from an auto-fluorescence reference (CHO Flp-In parental cell line) and 8-peak fluorescent calibration beads (BD, 653144). To normalize the gated, filtered fluorescence events, we applied the technical fluorescence variation control data to the following formula:

$$EGFP^i_{norm} = \frac{EGFP^i_{filtered} - EGFP_{auto}}{EGFP_{max}}, \qquad (2)$$

where $EGFP^i_{norm}$ is the normalized fluorescence from individual cells with an unsilenced circuit, $EGFP_{auto}$ is the mean autofluorescence from the parental cell line without the circuit, and $EGFP_{max}$ is the mean fluorescence of the highest fluorescence intensity peak from the calibration beads control.

Using each individual normalized fluorescence reading, we calculated the normalized mean and CV directly from the $EGFP^i_{norm}$ values using standard formulas: $CV = \sigma(EGFP^i_{norm})/\mu(EGFP^i_{norm})$, where $\mu(EGFP^i_{norm}) = \frac{1}{N}\sum_{1}^{N}EGFP^i_{norm}$ and $\sigma^2(EGFP^i_{norm}) = \frac{1}{N}\sum_{1}^{N}(EGFP^i_{norm})^2 - \mu^2(EGFP^i_{norm})$.

We plotted the mean and standard error of the mean (SEM) over replicates for each condition. In imaging, the number of cells per field was determined by spot detection of green-fluorescent cells and nuclei stained with the live cell dye NucBlue using NIS Elements AR (see Supplementary Methods for a detailed description). The non-parametric two-tailed Mann–Whitney U test inferred

significant differences at a $p$ value < 0.05. To calculate the adaptation time parameter, we fit the Baranyi environmental adjustment model[70] through minimization between the model and the data using Powell's method (Supplementary Fig. 38).

**Computational modeling**. To relate the fraction of mPF-PuroR and mNF-PuroR cells surviving the death phase of Puromycin treatment to *EGFP* and Puromycin concentrations, we first log transformed the approximately lognormal experimental CHO *EGFP* histograms, which were then fit to the Gaussian (normal) probability density function (Supplementary Fig. 17a):

$$f(EGFP_{mPF,mNF}) = \frac{1}{\sqrt{2\pi}\sigma_{mPF,mNF}} \exp\left(-\frac{(EGFP_{mPF,mNF} - \mu_{mPF,mNF})^2}{2\sigma_{mPF,mNF}^2}\right), \quad (3)$$

where *EGFP* is the level of the fluorescence reporter which corresponds to the expression level of the Puromycin resistance gene (*PuroR*), and $\mu$ and $\sigma$ are the mean and standard deviation of the population *EGFP* distribution for the mammalian positive or negative feedback circuit as indicated by the indices mPF and mNF, respectively.

The cumulative distribution function was directly obtained by integrating Equation (3) and is related to the fraction of cells that are initially killed by Puromycin ($A_D$)

$$A_{D_{mPF,mNF}}(EGFP) = \int_0^{A_D \text{thres}} \frac{1}{\sqrt{2\pi}\sigma_{mPF,mNF}} \exp\left(-\frac{(EGFP'_{mPF,mNF} - \mu_{mPF,mNF})^2}{2\sigma_{mPF,mNF}^2}\right) dEGFP'$$

$$(4)$$

The fraction of clonal CHO cells that initially survive Puromycin ($A_S$) is then simply $1-A_D$ (Supplementary Fig. 17b). The *EGFP* expression threshold below which CHO cells were killed is related to Puromycin concentration ([Puro]) via a Michaelis-Menten type function (Supplementary Fig. 17c):

$$A_{D_{\text{thres}}}(\text{Puro}) = \beta\left(\frac{[\text{Puro}]}{K + [\text{Puro}]}\right). \quad (5)$$

The fraction of initial surviving cells can be further divided into a persister cell fraction ($A_P$) and a nongenetically resistant fraction ($A_N$) such that $A_D + A_P + A_N = 1$. The $A_P$ fraction was estimated using a lognormal distribution function (Supplementary Fig. 17d):

$$A_{P_{mPF,mNF}}([\text{Puro}]) = \frac{1}{\sqrt{2\pi}\sigma'_{mPF,mNF}} \exp\left(\frac{-(\ln([\text{Puro}]) - \mu'_{mPF,mNF})^2}{2\sigma'^2_{mPF,mNF}}\right). \quad (6)$$

The initial subpopulation fractions of $A_D$, $A_P$, and $A_N$ served as input to a stochastic population dynamics model [Equation (7)], which accounted for the phenotype switching between $P$ and $N$ cells and the mutation of $N$ or $P$ cells to form a genetically drug-resistant subpopulation fraction of $G$ cells ($A_G$), which like $N$ cells could grow and divide. The model predicted subpopulation fractions and adaptation (Fig. 6):

$$\begin{aligned}
\frac{dP}{dt} &= r_{P,N}N - r_{N,P}P - r_{G,P}P \\
\frac{dN}{dt} &= -r_{P,N}N + r_{N,P}P - r_{G,N}N + k_N N - g_N N, \\
\frac{dG}{dt} &= r_{G,P}P + r_{G,N}N + k_G G - g_G G
\end{aligned} \quad (7)$$

where $r_{i,j}$ is transition rate from genotype/phenotype j to i, $k_i$ is the growth rate of i, and $g_i$ is the death rate of i. The total population size $N_{\text{tot}}$ is given by:

$$\frac{dN_{\text{tot}}}{dt} = \frac{dP}{dt} + \frac{dN}{dt} + \frac{dG}{dt} = k_N N + k_G G - g_N N - g_G G. \quad (8)$$

See Supplementary Methods for details on the implementation of the computational model. All parameter values are given in Supplementary Table 2.

**Puromycin treatment phase**. Prior to experimental design, a kill curve for ancestral CHO cells evaluated the minimal Puromycin concentration affecting cells without the resistance gene (see Supplementary Methods). In the initial drug treatment experiment, cells were seeded at $5 \times 10^4$ cells in 6-well plates and incubated 24 h prior to Doxycycline induction. After 48 h of induction, 8,000 cells were split in replicates of six into 24-well plates and induced for another 24 h before treatment with Puromycin. At 72 h, expression was measured by flow cytometry to determine the existence of a decoupled noise point between the two circuits, with the criteria being means that differ by less than 10%. Once the decoupled noise point was established, the cells were treated with varying levels of Puromycin. Plates were replenished with media, Doxycycline, Puromycin, and NucBlue every 24 h before imaging (Fig. 5a). If a well became confluent during the first treatment phase, we ran the sample under flow cytometry, temporarily removed Puromycin, split the sample into two new wells with or without Doxycycline, and the remaining cells were cryogenically frozen (Fig. 7a).

**Adaptation criteria**. We employed custom MATLAB scripts to estimate the local slope (growth rate) of the growth curves with a moving window of 3 timepoints (Supplementary Figs. 10; 34c, d). If the local growth rates up to saturation of the

growth curve were equal to or greater than 0, then the adaptation time = 0. If any local growth rate of a curve was less than zero, then we extracted the half-saturation time and adaptation time from the Baranyi model. Replicate populations that completely die off have adaptation times = infinity.

**Post-treatment and Re-treatment phases**. Samples that survived the Puromycin treatment phase were separated into two conditions: i) no Doxycycline and no Puromycin; and ii) with Doxycycline and without Puromycin. To maintain the same concentration of Doxycycline between passages, we neutralized 100 μL of trypsinized solution with 1 mL of either 0.055 or 6.6 ng/mL Doxycycline to dilute the solutions to 0.05 and 6 ng/mL, respectively. The media was replenished with the appropriate induction levels after adherence. Expression was monitored by flow cytometry during each passage. The number of passages for the samples required to reach the re-treatment phase varied between two to nine. If the expression completely reset, the mean expression for the uninduced sample reached lower levels than induced cells over a substantial amount of time (weeks), or the uninduced sample mean expression levels did not change over a month, the samples were retreated with the previously used Puromycin concentration. All lineages were cryogenically preserved between each passage and after the re-treatment phase. Additionally, uninduced mNF-PuroR or mPF-PuroR cells were measured by flow cytometry at each passage as a positive control for a successful reset.

**Genomic DNA extraction and sequencing analysis**. Ancestral and evolved CHO populations were centrifuged for 5 min at $300 \times g$, and the genomic DNA from the cell pellet was either extracted with the DNeasy Blood & Tissue kit (QIAGEN, 69504) or immediately frozen at −80 Celsius for future extraction. For all samples except mNF-PuroR replicates 1, 3, 4, 5, and 6, the entire sample was immediately centrifuged after thawing from cryopreservation. Otherwise, one-tenth of the sample was grown to confluence up to a 6-well plate, which was then lysed for DNA extraction. Once purified, the genomic DNA acted as a template for PCR amplification of the mNF-PuroR or mPF-PuroR circuits using primers listed in Supplementary Table 4. Various sequencing primers were used for Sanger sequencing such that the chromatographs covered the circuits with at least 2 reads. The ab1 files were aligned and peaks visualized with the SnapGene software (from GSL Biotech; available at snapgene.com). To assess for genetic heterogeneity in a sequencing read with peak mixtures, CRISP-ID[58] analyzed individual chromatographs for variant subpopulation sequences (see Supplementary Methods).

**Reporting Summary**. Further information on research design is available in the Nature Research Reporting Summary linked to this article.

## Data availability

The raw growth curve data for Figs. 5c–g, 7e–g; DNA sequences for the plasmids shown in Figs. 2a, 3a, Supplementary Figs. 4a, 6a; ab1 sequencing traces with associated template DNA sequence files for Supplementary Figs. 23–26, 29–31; and flow cytometry data for Figs. 2b–d, 3b–d, 4a–c, 7b–d, and associated Supplementary figures can be found at https://openwetware.org/wiki/CHIP:Data. The remaining data supporting the findings in the study are available from the corresponding author, G.B., upon reasonable request. Source data are included as a separate source file.

## Code availability

The MATLAB codes used to generate the computational results reported in this study are available at: https://github.com/dacharle42/MDR.

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

## Acknowledgements

This research was supported by the National Institutes of Health Director's New Innovator Award Program 1DP2 OD006481-01, NIGMS 1R01GM106027-01, NIGMS

1R35GM122561-01, and the Laufer Center for Physical and Quantitative Biology. Sequencing was performed at MD Anderson's DNA Analysis core facility (funded by NCI CA16672) and Stony Brook School of Medicine's DNA Sequencing Core facility. Flow sorting of polyclonal transfections was done at the Stony Brook School of Medicine's Research Flow Core facility. DN acknowledges support from program # 1326, the Ministry of Education and Science, Russian Federation. We would like to thank all Balázsi lab members for discussions, Lucie Chrastecka for help with the experiments requiring daily attention, Christopher Balzano for advice regarding short-term storage of Doxycycline solutions containing serum, and M. Tyler Guinn for help with Adobe Illustrator.

## Author contributions

K.F. and G.B. designed the research; D.N. designed and built the mNF-GFP gene circuit; K.F. built all other gene circuits, K.F. performed the majority of experiments with contributions by M.S. and J.C.; D.C. and G.B. performed the mathematical modeling; D.C. designed and performed the computational simulations; K.F., M.S., J.C. and G.B. analyzed the data, and K.F., D.C. and G.B. wrote the paper.

## Additional information

**Competing interests:** The authors declare no competing interests.

