## [Peer Review File · Nature Communications]

Reviewers' comments:

Reviewer #1 (Remarks to the Author):

The study by Farquhar et al. aims to demonstrate the role of gene expression noise in the development of drug resistance in mammalian cells. This is an important and timely pursuit. Gene expression noise has been related to drug resistance in other cell types where increased variability has been shown to be beneficial for resisting high levels of drug treatment, and it has been long theorized that something similar should occur in mammalian cells with important potential implications for the failure of certain drug therapies in certain cancer patients.

To explore how gene expression noise impact the emergence of drug resistance, the authors integrate two simple synthetic circuits to decouple mean gene expression from variability. They authors effectively show using CHO cells that at the same mean expression level, high noise is enhances resistance under high levels of drug treatment, while the opposite is true under low levels of drug treatment. These experimental results are supported by computational models purportedly based on experimental data. These models are in turn used to predict that the mechanisms underlying drug resistance are likely genetic.

The lasting effects of potential genetically-coded resistance are investigated by re-treating adapted cells after a period of drug-free growth. The overall approach of experimentation informing computational models which in turn informed further experiments to confirm the models was very coherent and each part supported the others nicely.

Overall, the work clearly demonstrates that gene expression noise is beneficial for drug resistance at high levels, but not at low levels, of stress. The use of synthetic circuits to simplify the system strongly supports that the observed differences really are due to differences in gene expression noise alone. A reasonable mechanism of permanently increased resistance through genetic mutation is proposed and then tested experimentally as well.

I think it is a well-designed study and the findings are important. Nonetheless, I am not entirely convinced that the interpretation of the experimental data is as strong as it could or should be. Similar results have been reported for experiments with yeast cells and that data was explained using mathematical models without permanent and inheritable genotype alterations. To my knowledge, gene expression variability can have a broad spectrum of time-scales and it is not obvious to me how slowly evolving lineage-specific effects can be ruled out conclusively without identifying and validating the alternative hypothesis that the increased resistance is due to genetic mutations. The identification and validation of such mutations would also rule out other potential modes of drug resistance development.

Minor notes

It is mentioned that a similar PF system in yeast produced bimodal gene expression patterns whereas expression is unimodal in mammalian cells. I think this would be very interesting to explore, or at the least provide some reasonable explanation for why this might be.

The division of the major growth experiments into "Experiment 1" and "Experiment 2" was a little confusion as they were essentially the same experiments but using different concentrations of puromycin. As each of these "experiments" consisted of multiple concentrations of puromycin anyways, I don't see why they can't be grouped as one single set of experiments with increasing levels of puromycin.

It is mentioned that the computational models based on the initial experiments predict a beneficial mutation rate of $\sim 10^{-5}$ /genome/hour. Is this mutation rate physiologically possible for non-growing persister cells? If not, perhaps there is a different mechanism.

Tables S3a-c are not very intuitive or immediately clear. What each graph represents in terms of cell number, gene expression, and drug addition/removal/re-addition is not clear from the tables. Maybe labels can be added to the tables or in the legend. The reason why certain scenarios are excluded (red rectangles) is also not immediately clear, maybe each rectangle can be labelled with the figure to which it corresponds.

Reviewer #2 (Remarks to the Author):

If noise in gene expression can help or hinder adaptation and evolution is an important question. The authors make some progress here using two constructs that have the same expression but different noise, and measuring drug resistance.

We have three major concerns.

(1) It seems impossible to differentiate the effect of noise from the difference in network architecture. All of the effects may be due to the positive feedback loop, and not the high noise.

(A) High expressing cells with the PF circuit may maintain high expression, and this could generate the increased survival at high drug concentration.

There are a few ways to solve this problem: (1) do the drug resistance experiments at an expression level of 1 (Fig 4a) and see that the two circuits behave the same, then this may result in too high expression of PuroR to see any difference. (2) Construct a high noise circuit without PF (this looks promising: Mundt et al. bioRxiv 2018. A system for gene expression noise control in yeast). (3) Sort the high expressing PF & NF cells and show that they revert to the mean expression at the same rate, but the data suggest that this will not occur -- growth in puromycin will likely continuously select for the high expression PF cells.

It is possible that we missed a key point of evidence that differentiates noise from circuit architecture -- the figures are generally difficult to understand.

Network vs noise has to be differentiated if we are to understand the data.

(B) The different mutation rates (if that is indeed occurring) may be due to the PF circuit being more sensitive to mutations, and have nothing to do with high noise.

(C) There may be no difference in mutation rate -- the PF circuit may maintain high expression epigenetically.

(2) The claim that regrowth rates indicate that noise aids evolutionary adaptation in high stress, while the reverse is true for low stress - is weak. The increase in regrowth rate in the NF strain is of similar size between experiment 1 & 2 moving from 22->35ng/ul as it is between NF & PF at 10ng/ul. Get better data for regrowth rate (more drug concentrations), display the data in a more convincing manner, or de-emphasize the difference in regrowth rate.

Why is growth rate higher for NF at 10ug/ml, but lower for NF at 22ug/ml? Non-monotonicity without a reasonable explanation suggests that this result is not real.

(3) In the model, the authors assume that non-genetically drug resistant cells can not mutate. We don't understand why. Why not NR->GR? This has to be explained.

Minor points:

The authors suggest that in low noise system resistance is mainly epigenetic whereas in high noise quite often genetic (under the ~medium stress, at the concentration 22.5ug/ml). Moreover, the adaptation time is much shorter for NF than for PF. Any follow up suggestions on the origin of the difference? In discussion maybe? Proposals.

The authors measure noise in two systems, providing positive and negative feedbacks. In negative feedback they claim that the distribution is very broad but yet unimodal. Be very careful with the word unimodal because it easily can be that distribution has two stable subpopulations but the mean of both is close resulting in the merging of two modes. It's not important, but I would recommend not to claim unimodality in a broad distribution. "not visibly bimodal" or something like that.

We understand the reason for the break between exps 1 & 2 (eg: Fig 5i) but this is confusing and needs to be made more clear in the figure. eg: dashed line for exp 1 & solid for exp 2 throughout all figures in the paper.

"suggesting that their characteristics would also vary similarly" -- I don't understand what this means.

It is generally difficult to compare expression values across figures. Make it more obvious that 4b is a zoom of 4a. Is the expression comparable between Fig4 & Fig7? Do the cells in fig 7 really have 10 fold higher expression?

This seems like selection for a very high expressing subpopulation, which can be maintained by PF, as opposed to mutation.

The figures, and axis labels, are generally difficult to understand.

Overall we enjoyed the paper, and find results interesting, but have some questions and doubts regarding the interpretation of the data.

Lucas Carey
lucas.carey@upf.edu

Alsu Missarova
alsu.missarova.32@gmail.com

Reviewer #3 (Remarks to the Author):

Farquhar et al demonstrate the underlying impact of gene expression noise in the context of mammalian cell drug resistance evolution by decoupling noise from mean gene expression. The authors achieve this by comparing two cell lines each with an inducible synthetic gene circuits, the first a positive feedback system that confers antibiotic resistance to puromycin with high gene expression noise, and the second synthetic gene circuit conferring antibiotic resistance with low noise. Titration of the doxycycline inducer enables the two circuits to achieve similar gene expression means, while the nature of the positive/negative feedback systems impacts the noise of their respective circuits. The authors found that gene expression noise can either aid or hinder adaptation depending on the level of stress (antibiotic concentration).

The findings are interesting and novel for mammalian synthetic biology, yet there are a few concerns that ought to be addressed before it can be recommended for publication.

1. How do mNF and mPF work exactly? The authors may want to briefly discuss the operation of these systems. Does the mPF system rely on leakage?
2. The authors suggest that there are genetic adaptations that influence PuroR expression, but there is no sequencing information. Given that this is a synthetic circuit, it should be easy to detect genetic changes via sequencing.

3. Epigenetic alterations may be influential (as the authors suggest), but if the cassettes were integrated in the same locus between cell lines, this may be less of a concern. Verifying the location is essential.
4. The authors use EGFP as a proxy for PuroR expression, it would be advisable to perform qPCR of PuroR and EGFP between the cell lines at the chosen doxycycline concentrations in order to demonstrate this relationship.
5. Figure 5a is unclear in the experimental set-up. Additionally, there seem to be many microscopy images not submitted in the report. This should at least be included in the supplement.
6. In multiple figures, there are abbreviations that are not explained in the text.
7. A number of papers discuss synthetic microRNA-based feedforward loops and should at least be cited in the introduction/discussion: PMID 21811230, 21423718, 29354284.
8. The authors integrated their cassettes into what they describe as “the well-expressed Flp-In locus”. I believe that the term “Flp-In locus” is not accurate. These are cell lines that have been derived by inserting the FRT site in random locations, typically then selecting clones that are transcriptionally active and have low epigenetic silencing.
9. Did the authors start with a monoclonal Flp-In cell line? What is the actual location of integration? Were the circuits integrated as a single copy? The authors have to verify the location and copies.

RESPONSE TO THE REVIEWERS

We would like to thank the Reviewers for the critical reading of our manuscript and for their helpful comments. We were delighted that each Reviewer appreciated our study, considering it to be “important and timely”, addressing an “important question”, and “interesting and novel for mammalian synthetic biology”. Motivated by the Reviewers’ comments, we performed additional experiments and computational modeling, which led to new conclusions, thereby strengthening the manuscript. We have rewritten the manuscript substantially to address the Reviewers’ concerns and incorporate the new findings. We hope that the revised manuscript is sufficiently convincing, making it acceptable for publication. Below, we respond point-by-point to each Reviewer’s comments.

REVIEWER #1

Major concern:

Nonetheless, I am not entirely convinced that the interpretation of the experimental data is as strong as it could or should be. Similar results have been reported for experiments with yeast cells and that data was explained using mathematical models without permanent and inheritable genotype alterations. To my knowledge, gene expression variability can have a broad spectrum of time-scales and it is not obvious to me how slowly evolving lineage-specific effects can be ruled out conclusively without identifying and validating the alternative hypothesis that the increased resistance is due to genetic mutations.

Response: Thank you for this excellent and important comment. We agree with this possibility. To examine the timescales of noise (gene expression fluctuations) for both gene circuits, we estimated the memory of high-noise and low-noise cells by flow-sorting high- and low-expressing subpopulations and monitoring their expression over time. We found that cells with the high-noise mPF-PuroR gene circuit have higher memory (~2 days) of both high and low gene expression states compared to the low-noise gene circuit (~1/2 day; **Supplementary Fig. S7; Supplementary Table 1**). This is expected due to the positive feedback architecture, which can amplify fluctuations and increase fluctuation relaxation times¹, while negative feedback tends to have the opposite effect. Therefore, in principle, slow mPF relaxation time to the equilibrium distribution could maintain mPF cells at expression levels sufficiently high for resistance, such that they can survive and grow up during drug treatment. However, if fluctuations with experimentally measured timescales were the sole sources of drug resistance, without any persister-type cell cycle arrest, the surviving cells should start visibly growing in a few days instead of multiple weeks as observed experimentally. Indeed, computational models of gene expression using Ornstein-Uhlenbeck processes with relaxation times matching those of the gene circuits failed to reproduce the experimental evolutionary dynamics, specifically the long delays without growth followed by fast regrowth at high stress. Therefore, the different pre-treatment fluctuation timescales and noise amplitudes might very well account for short-term survival, but they alone cannot explain the different long-term evolutionary dynamics of the gene circuits (**Supplementary Fig. S16**). Admittedly, sequencing could not rule out the contribution of epigenetic alterations to drug resistance in most replicates. Thus, if phenotypic switching to a persister-like state and stable epigenetic alterations can be considered “slowly evolving lineage-specific effects” then such effects are entirely possible and consistent with our results.

The identification and validation of such mutations would also rule out other potential modes of drug resistance development.

Response: We agree. To address this comment and determine whether genetic mutations contributed to drug resistance, we sequenced both gene circuits after adaptation to 22.5, 35, and 50 µg/mL Puromycin. In one replicate of the low-noise mNF-PuroR gene circuit we found an indel in the *hTetR* regulator gene near its DNA-binding domain (**Supplementary Fig. S22**) that can reduce TetR binding affinity to *tetO2* operator sites by 1000-fold². This explains the elevated expression and resistance of cells in this evolving replicate. We also found evidence for genetic heterogeneity in the distal-early CMV enhancer of the low-noise mNF-PuroR gene circuit in two other replicates (**Supplementary Figs. S23-24**). These latter mutations share the same single nucleotide change, which is absent in the ancestral gene circuit, and the phenotype of cells with these mutations is consistent with lost TetR function. Therefore, it is reasonable to conclude that these mNF-PuroR mutations also arose under selection for losing TetR function and thus elevating *PuroR* expression. Although we could not find any gene circuit mutations in the remaining three mNF-PuroR replicates (**Supplementary Fig. S25**), they were phenotypically identical (highly elevated, stable, inducer-independent *PuroR* expression) to the ones with gene circuit mutations. This is consistent with intra- and extra-circuit heritable alterations, all somehow abrogating TetR function. In contrast, we found no mutations in any evolved replicate of the high-noise mPF-PuroR gene circuit. Since *PuroR* expression in mPF-PuroR dropped after removal of induction (**Fig. 7c,d**), we hypothesize that adaptation occurs through heritable, genetic or epigenetic alterations outside of the gene circuit that confer induction-dependent, elevated *PuroR* expression. Overall, high noise and higher memory might aid short-term survival, but they alone seem insufficient to explain the long-term adaptation time course results unless we introduce a persister-like state that can later turn into stable drug resistance (**Supplementary Fig. S16**).

It is mentioned that a similar PF system in yeast produced bimodal gene expression patterns whereas expression is unimodal in mammalian cells. I think this would be very interesting to explore, or at the least provide some reasonable explanation for why this might be.

Response: Flow-sorting induced cells with the mPF-PuroR gene circuit revealed that the nongenetic memory of both up- and down-switching was ~2 days. Taken together with CHO growth rates of ~2/day, the switching rates and growth rates of CHO cells are relatively comparable. This sharply contrasts with yeast cells, whose growth rates of ~0.3/h are much faster than their memories of low- and high expression, ~24 hours and ~200 hours, respectively. Overall, yeast PF cells can divide many times before they switch, generating two clearly distinct peaks³. On the other hand, CHO cells with the mPF-PuroR gene circuit fluctuate and grow at comparable time scales, making their expression peaks (if they existed) indistinguishable. It is also possible that the mammalian mPF-PuroR gene circuit is not bistable. Despite having similar gene circuit architectures, the yeast and mammalian PF gene circuits are based on entirely different promoters and are in different host cells, any of which could explain the differences between the yeast and mammalian gene expression distributions.

The division of the major growth experiments into “Experiment 1” and “Experiment 2” was a little confusion as they were essentially the same experiments but using different concentrations of puromycin. As each of these “experiments” consisted of multiple concentrations of puromycin anyways, I don’t see why they can’t be grouped as one single set of experiments with increasing levels of puromycin.

Response: We appreciate this comment. We initially considered the experiments separately because the normalized mean expression and CV values differed slightly between the two experiments, even though we decoupled the noise from the mean in both experiments. Considering this, we decided to still maintain, but clarify this separation while plotting data from both experiments in the same figures. For example, the decoupled noise points for the experiments are now co-plotted in **Fig. 4b**. Additionally, we plot the growth curves for the first experiment with dash-dot lines as opposed to continuous lines for the second experiment in **Fig. 5c-h**.

It is mentioned that the computational models based on the initial experiments predict a beneficial mutation rate of $\sim 10^{-5}$ /genome/hour. Is this mutation rate physiologically possible for non-growing persister cells? If not, perhaps there is a different mechanism.

Response: Before answering this important question, we would like to clarify that: (i) the rate of random mutations is higher than the rate of beneficial mutations for any cell type in any condition; and (ii) often only random mutation rates are experimentally measured, so beneficial mutation rates are less well known for any cell type – and they can strongly depend on the environmental conditions. In the revised model, both persister-like cells and nongenetically resistant cells can mutate to become fast-growing genetically resistant cells. The reported range of the random CHO mutation rates is 10^{-7} to 10^{-3} per genome per generation⁴⁻⁷. Since we observed an experimental doubling time close to 13 hours based on the growth rates without treatment, the spontaneous, random mutation rate estimate of growing CHO cells ranges from $\sim 10^{-7}$ to 10^{-4} per genome per hour, which is overall consistent with the 10^{-7} to 10^{-6} per genome per hour beneficial mutation rate range used in our revised model. It is important to note that “persister” in our case is interpreted more broadly than its classical definition, simply as non-growing and non-dying cells by any mechanism. Recent evidence for stochastic phenotype switching⁸⁻¹¹ and mutation of slow-growing tolerant cells to become resistant to antibiotics^{12,13} supports the revised model, which was the only model that could capture the experimental adaptation times (**Fig. 6**). Interestingly, we observed a subpopulation of cells with multiple nuclei under 35 and 50 $\mu\text{g/mL}$ Puromycin. This phenotype has been reported to confer drug resistance in cancer¹⁴, and may be consistent with the broadly defined ‘persister-like’ cell state in our model.

Tables S3a-c are not very intuitive or immediately clear. What each graph represents in terms of cell number, gene expression, and drug addition/removal/re-addition is not clear from the tables. Maybe labels can be added to the tables or in the legend. The reason why certain scenarios are excluded (red rectangles) is also not immediately clear, maybe each rectangle can be labelled with the figure to which it corresponds.

Response: Thank you for pointing this out. We agree and have now completely removed the tables and reincorporated the logical points through flow charts of adaptation scenarios (**Supplementary Figs. S18; S26; S32; S34**). The scenarios categorize outcomes as either stochastic effects or heritable alterations which can be *PuroR*-dependent or based on native drug resistance mechanisms. The scenarios are further categorized by induction-dependence and cost of mutations. We ruled out various scenarios depending on the gene expression and adaptation time data during drug removal and retreatment, as explained in the legends.

REVIEWER #2

Major concerns:

(1) It seems impossible to differentiate the effect of noise from the different in network architecture. All of the effects may be due to the positive feedback loop, and not the high noise.

Response: Thank you for this comment, with which we agree completely. Indeed, the noise properties (noise amplitude, memory, etc.) of any gene are deeply intertwined with the gene's embedding regulatory network. While in the future it may be possible to tease apart the network from the noise, here we decided to simply admit that networks and noise are not trivially separable. Therefore, we rewrote the manuscript to focus on comparing the effects of high-noise, positive-feedback networks versus low-noise, negative feedback networks, without separating networks from their noise properties. We still consider as an interesting basic problem to compare the evolution of high-noise mPF networks with those of low-noise mNF networks, starting them in the decoupled noise regime. "Decoupled noise" now means that the gene expression means are similar while everything else (noise amplitude, memory, network) can be different. We think that the phenomena we observe occur in two phases, corresponding to two different timescales: (i) initially, the actual network is probably less relevant than its noise properties (e.g., amplitude and memory), which mediate short-term survival immediately after drug treatment; (ii) later, at long-term evolutionary time scales, the network topology becomes more important as it constrains the network modification variants that can evolve while the cells gain drug resistance. We admit that positive feedback is one mechanism that can slow fluctuation relaxation times^{1,15,16}, and increase noise amplitude, which we confirm with flow cytometry, flow-sorting and simulations (see **Supplementary Figs. S7; S16; Supplementary Table S1**). However, we suspect that other mechanism (for example, slow chromatin dynamics) and network topologies (such as certain feedforward loops) that confer similar noise properties would have similar effects on initial, short-term survival. On the other hand, the long-term evolution probably depends on the actual network topology, which determines the types of beneficial mutations that can confer the evolved drug resistance phenotypes.

(A) High expressing cells with the PF circuit may maintain high expression, and this could generate the increased survival at high drug concentration.

Response: We agree with this possibility and thank you for the comment. We have addressed a similar comment by Reviewer #1. To examine how long each gene circuit maintains its expression away from the mean, we estimated the memory of high-noise and low-noise cells by flow-sorting high- and low-expressing subpopulations and monitoring their expression over time. We found that cells with the high-noise mPF-PuroR gene circuit have higher memory (~2 days) compared to the low-noise gene circuit (~1/2 day; **Supplementary Fig. S7; Supplementary Table 1**). This is expected due to the positive feedback architecture, which can amplify fluctuations and increase fluctuation relaxation times¹, while negative feedback tends to have the opposite effect. Therefore, in principle, slow relaxation time to the equilibrium distribution could maintain mPF cells at higher expression levels relative to the low-noise mNF cells, enabling growth during drug treatment. However, if slow fluctuations were the sole source of drug resistance, then mPF should always prevail over mNF, which is not the case. Also, if slow fluctuations would underlie drug resistance without any phenotypic switching to a cell cycle arrested persister-state, the surviving cells should start visibly growing in a few days, as opposed to the experimentally observed multi-week delays without growth. Indeed, computational models with Ornstein-Uhlenbeck processes with noise amplitudes, means and

relaxation times matching those of the gene circuits failed to reproduce the experimental evolutionary dynamics, specifically the long delays without growth, followed by fast regrowth at high stress. In summary, the different fluctuation timescales and noise amplitudes of nongenetic processes might very well account for short-term survival, but they alone cannot explain the different long-term evolutionary dynamics of the gene circuits in our experiments (**Supplementary Fig. S16**).

There are a few ways to solve this problem: (1) do the drug resistance experiments at an expression level of 1 (Fig 4a) and see that the two circuits behave the same, the this may result in too high expression of PuroR to see any difference.

Response: We appreciate this comment. However, as the Reviewer mentioned, a *PuroR* expression level of 1 is too high to reasonably recapture the adaptation curves since the current Puromycin doses would be practically ineffective. Finding and using much higher Puromycin doses would be time-consuming and problematic. Finally, the time necessary for setting up decoupled noise control and repeating the evolution experiments again would be unreasonably long for this resubmission.

(2) Construct a high noise circuit without PF (this looks promising: Mundt et al. bioRxiv 2018. A system for gene expression noise control in yeast).

Response: This is indeed an interesting suggestion, which we would like to consider in future studies. However, the time needed to construct and test this new system for decoupled noise control in mammalian cells and then repeating the evolution experiments would be unreasonably long for this resubmission.

(3) Sort the high expressing PF & NF cells and show that they revert to the mean expression at the same rate, but the data suggest that this will not occur -- growth in puromycin will likely continuously select for the high expression PF cells.

It is possible that we missed a key point of evidence that differentiates noise from circuit architecture -- the figures are generally difficult to understand.

Network vs noise has to be differentiated if we are to understand the data.

Response: We agree with this excellent and feasible suggestion, and the prediction that mNF and mPF cells do not revert to the mean expression at the same rate. Accordingly, we flow-sorted mPF and mNF cells into high- and low-expressing subpopulations and monitored expression over time. We fitted the high-expressing fraction of cells in each sorted expression subpopulation to an analytical solution for a two-state gene expression model¹¹. The solution indicated that mPF cells have up- and down-switching rates of ~2 days (**Supplementary Fig. 7; Supplementary Table 1**). The mNF cells switched back faster to their original distribution, at rates of ~1/2 day. Thus, fluctuations back to the equilibrium distributions indeed depend on the network: cells with the higher noise-amplitude also have higher noise-memory. Nonetheless, as we discussed above, these differences in noise and memory are deeply intertwined and are not trivially separable from the network topology. Moreover, these noise and memory differences cannot explain the experimental adaptation dynamics. Therefore, in the rewritten manuscript we do not separate networks from their noise properties. Rather, we compare the evolutionary effects of a high-noise mPF network with a low-noise mNF network, keeping the networks and their noise properties convoluted. We think the basic problem of comparing the evolution of high-noise mPF networks with those of low-noise mNF networks, starting them in the decoupled noise regime is still interesting. "Decoupled noise" now means that the gene expression means are similar while everything else (noise amplitude, memory, network) can be different. Finally,

while the noise properties, including memory differences could explain short-term survival, they alone cannot explain long-term adaptation in our experiments (**Supplementary Fig. S16**). So, although continuous selection of high expression mPF cells does occur, the surviving/selected cells enter a persister-like state and later mutate, instead of growing immediately.

(B) The different mutation rates (if that is indeed occurring) may be due to the PF circuit being more sensitive to mutations, and have nothing to do with high noise.

Response: Thank you for this excellent comment. We removed some of these statements from the manuscript. The beneficial mutation rates indeed reflect the network's mutational capacity to elevate *PuroR* expression and have nothing to do with noise, except that the same network underlies both the noise and the mutational possibilities. We discuss this in the current version of the manuscript, distinguishing two separate time scales: (i) short-term survival when fluctuating *PuroR* levels determine whether a cell survives or not while the network topology is relatively "hidden"; versus (ii) long-term evolution when the cells gain resistance by altering the network topology, which thereby becomes directly relevant. We estimated the mutation rate computationally. Determining the mutation rate experimentally would be very difficult and out of scope, since the heritable alterations are often outside of the gene circuit and could be genetic or epigenetic. So, we sought to identify intra-circuit mutations by Sanger-sequencing the gene circuit in many samples. Interestingly, we found no mutations or evidence of genetic heterogeneity in the high-noise mPF gene circuit in cells treated under 22.5, 35, and 50 $\mu\text{g}/\text{mL}$ Puromycin. On the other hand, we did find evidence of genetic mutations in 3 replicates of the mNF gene circuit under 35 $\mu\text{g}/\text{mL}$ Puromycin. Specifically, mNF replicate 2 has a fixed 6-bp indel that deletes Glycine 22, which was found to reduce binding affinity of the TetR repressor to target *tetO2* sites in the promoter by 1000-fold² (**Supplementary Fig. S22**). We also found evidence of genetic heterogeneity in the mNF gene circuit promoter from replicates 1 and 3 by distinguishing alleles in sequencing trace mixed peaks using CRISP-ID¹⁷ (**Supplementary Figs. S23-24**). The other mNF replicates do not contain mutations (**Supplementary Fig. S25**) but share the same phenotype of highly elevated, inducer-independent *PuroR* expression. Thus, we concluded that adaptation also occurred through extra-circuit heritable mechanisms that abrogate TetR repressor function (see the adaptation flow charts).

Overall, the mNF gene circuit seems "more sensitive" to mutations, which is interesting yet expected. Functionally deleterious mutations occur easier in the mNF gene circuit than the mPF gene circuit probably because any possible way of breaking the repression mediated by the TetR protein allows for increased *PuroR* expression. On the other hand, elevating *PuroR* expression for the mPF gene circuit would require mutations in *rtTA* that increase its ability to activate transcription. This would amount to improved *rtTA* protein function, which is more difficult to achieve mutationally than broken TetR protein function. The current focus of the manuscript on high-noise versus low-noise networks creates room for all these possibilities.

(C) There may be no difference in mutation rate -- the PF circuit may maintain high expression epigenetically.

Response: We agree. Now we make no statements about overall beneficial mutation rates. However, we did find differences in intra-circuit mutation rates. The high-noise mPF network did not mutate at any treatment level, while three adapted low-noise mNF replicates contained mutations in the gene circuit. We discussed the reasons for this above (see response to B). Nonetheless, evidence exists for extra-circuit heritable alterations in cells with the high-noise mPF gene circuit at high stress. We have also discussed that the maintenance and selection of high mPF expression is insufficient to explain the experimental findings (see response to A).

(2) The claim that regrowth rates indicate that noise aids evolutionary adaptation in high stress, while is reverse true for low stress - is weak. The increase in regrowth rate in the NF strain is of similar size between experiment 1 & 2 moving from 22->35ng/ul as it is between NF & PF at 10ng/ul. Get better data for regrowth rate (more drug concentrations), display the data in a more convincing manner, or de-emphasize the difference in regrowth rate.

Why is growth rate higher for NF at 10ug/ml, but lower for NF at 22ug/ml? Non-monotonicity without a reasonable explanation suggests that this result is not real.

Response: Thank you for this comment. We agree. Since adaptation times are sufficiently different and more relevant for the evolutionary effects of noisy networks, we have de-emphasized the regrowth rates and focused on adaptation times instead in the revised manuscript.

(3) In the model, the authors assume that non-genetically drug resistant cells can not mutate. We don't understand why. Why not NR->GR? This has to be explained.

Response: Thank you for this great comment. We have revised the computational models to include this transition (**Fig. 6**). The revised models incorporate NR->GR mutation, in addition to P<->NR phenotype switching, and P->GR mutations. We used parameter scans to test the importance of each process for agreement with the experimental adaptation-time results.

Minor points:

The authors suggest that in low noise system resistance is mainly epigenetic whereas in high noise quite often genetic (under the ~medium stress, at the concentration 22.5ug/ml). Moreover, the adaptation time is much shorter for NF than for PF. Any follow up suggestions on the origin of the difference? In discussion maybe?
Proposals.

Response: Based on the new sequencing results, we have changed these conclusions. In fact, mNF cells do not adapt under the 22.5 µg/ml Puromycin condition; the mNF cells do not have an initial delay before regrowth (**Supplementary Fig. 33c**) and some of their expression levels are likely above the threshold needed for survival. Thus, the mechanism in 22.5 µg/ml Puromycin is purely nongenetic: PuroR expression is sufficiently high in some mNF cells that they can grow. There is no need for stably genetic or “epigenetic” (meaning DNA methylation or alterations in chromatin accessibility, etc.) to explain these findings. The mPF cells take longer to adapt at this moderate stress level possibly because of the known harmful effect of high noise at low stress, potentially coupled with some cell population-wide response to enter the persister state. Based on the elevated mPF expression levels after drug removal and resistance during retreatment, extra-circuit heritable genetic or “epigenetic” alteration(s) likely contribute to inducer-dependent resistance in the high-noise mPF gene circuit under 22.5 µg/ml Puromycin. For 35 µg/ml Puromycin, both mNF and mPF surviving cells massively enter the persister state, and the adaptation times (delays) probably reflect the availability of beneficial mutations, which may be higher for mNF for reasons discussed above. Finally, for 50 µg/ml Puromycin the mNF cells most likely die before a beneficial mutation could arise (even if it is more likely). At this most severe stress level, higher noise-amplitude and noise-memory saves enough mPF cells and “buys” sufficient time for them to acquire beneficial mutations and grow.

The authors measure noise in two systems, providing positive and negative feedbacks. In positive feedback they claim that the distribution is very broad but yet unimodal. Be very careful

with the word unimodal because it easily can be that distribution has two stable subpopulations but the mean of both is close resulting in the merging of two modes. It's not important, but I would recommend not to claim unimodality in a broad distribution. "not visibly bimodal" or something like that.

Response: Thank you for the comment, which is reasonable. Indeed, we are not in a position to make claims about either bimodality or bistability, or lack thereof based on the distributions. Therefore, we have toned down the statements on the distributions being "unimodal".

We understand the reason for the break between exps 1 & 2 (eg: Fig 5i) but this is confusing and needs to be made more clear in the figure. eg: dashed line for exp 1 & solid for exp 2 throughout all figures in the paper.

Response: Thank you for this comment. We re-plotted the growth curves from the first experiments as dash-dot lines. We added dashed lines between the 22.5 and 35 $\mu\text{g/ml}$ Puromycin adaptation times to distinguish each experiment. This has hopefully improved the clarity of figures.

"suggesting that their characteristics would also vary similarly" -- I don't understand what this means.

Response: This has now been removed from the text.

It is generally difficult to compare expression values across figures. Make it more obvious that 4b is a zoom of 4a.

Response: Thank you for this suggestion - we have now added a black bracket in **Fig. 4a** that represents the expression range in **Fig. 4b**.

Is the expression comparable between Fig4 & Fig7? Do the cells in fig 7 really have 10 fold higher expression?

Response: Yes, the expression is comparable, which we ensured by fluorescence data normalization. There is really a 10-fold difference in expression. In fact, the raw (unnormalized) expression levels are also 10-fold higher (**Supplementary Fig. S19**).

This seems like selection for a very high expressing subpopulation, which can be maintained by PF, as opposed to mutation.

Response: We think that we have addressed this by answering the previous comments.

The figures, and axis labels, are generally difficult to understand.

Response: We have improved the figures and axis labels to address this concern.

REVIEWER #3

1. How do mNF and mPF work exactly? The authors may want to briefly discuss the operation of these systems.

Response: Thank you for this clarifying question. We have now included a description with references in the sections characterizing the mNF and mPF gene circuits, so it is more easily understandable how they function and how feedback relates to noise. Additionally, we redrew a simple diagram of negative and positive feedback in **Fig. 1** next to the distributions.

Does the mPF system rely on leakage?

Response: Both gene circuits have leaky expression compared to parental cells, with mNF having a higher level of basal expression (data not shown). In mPF, leakage is required to provide free *rtTA* for Doxycycline binding; otherwise expression of this gene circuit could not increase during induction.

2. The authors suggest that there are genetic adaptations that influence PuroR expression, but there is no sequencing information. Given that this is a synthetic circuit, it should be easy to detect genetic changes via sequencing.

Response: Based on the Reviewers' comments, we performed Sanger sequencing of the mNF-PuroR and mPF-PuroR gene circuits for all replicates in most experimental conditions. We found mutations in the *hTetR* repressor from the mNF-PuroR gene circuit in replicate 2 under 35 µg/mL Puromycin that previously have been reported to reduce affinity to *tetO2* sites in the promoter². Also, mNF replicates 1 and 3 exhibited signs of genetic heterogeneity in the promoter, which co-occur with highly elevated, inducer-independent *PuroR* expression, matching exactly the phenotype of replicate 2 with lost TetR function. The remaining mNF replicates had the same phenotype without any detectable gene circuit mutations, suggesting additional, extra-circuit mechanisms that abrogate TetR repressor function. There were no detectable mutations in the mPF-PuroR gene circuits evolved under any Puromycin level. We argue that extra-circuit heritable genetic or "epigenetic" (such as DNA methylation or chromatin modifications) mechanisms of resistance prevail for high-noise gene circuits, but investigating these would be difficult and beyond the scope of this study.

3. Epigenetic alterations may be influential (as the authors suggest), but if the cassettes were integrated in the same locus between cell lines, this may be less of a concern. Verifying the location is essential.

Response: The genomic location of the FRT sites is not publicly available from Invitrogen, probably for proprietary purposes. We have contacted the company, and they denied any knowledge of the location. Location mapping in-house would not be trivial, so we abandoned that goal. Nonetheless, both gene circuits were integrated into the same clonal CHO-Flp-In parental cell line. Therefore, the location of integration should be identical. Moreover, we determined through qPCR that each gene circuit is integrated as a single copy (**Supplementary Fig. S8**), and selection with Hygromycin B requires integration of the start-codon-less HygroB resistance gene into the genomic FRT site containing a start codon as a promoter trap (see the Methods). Thus, we conclude that the gene circuits are located in single copies at the same genomic FRT locus.

4. The authors use EGFP as a proxy for PuroR expression, it would be advisable to perform qPCR of PuroR and EGFP between the cell lines at the chosen doxycycline concentrations in order to demonstrate this relationship.

Response: EGFP and PuroR expression should be present in the same mRNA with equal stoichiometries. Regarding qRT-PCR, we think that the probes bind to the same tricistronic mRNA, which should indicate comparable expression levels (being the same mRNA molecule). Below we are showing data for another construct in our lab, with qRT-PCR probes for the different cistrons, demonstrating strong correlation between the two cistrons. Also, the literature seems to suggest that P2A-separated proteins have proportional levels of expression¹⁹, and P2A-separated reporters are used as proxies for stem cell regulators²⁰. Therefore, we think it is reasonable to assume that the protein levels translated from the same mRNA should be proportional to each other (even if not identical).

5. Figure 5a is unclear in the experimental set-up. Additionally, there seem to be many microscopy images not submitted in the report. This should at least be included in the supplement.

Response: We improved this figure by adding more detailed and clear diagrams, showing exactly what happened rather than a representation. We also added microscopy images to the SI (**Supplementary Figs. S12-15**), which highlights morphological diversity such as multi-nucleated cells.

6. In multiple figures, there are abbreviations that are not explained in the text.

Response: All abbreviations are now defined upon initial use in the main text and SI and defined again in the figure captions.

7. A number of papers discuss synthetic microRNA-based feedforward loops and should at least be cited in the introduction/discussion: PMID 21811230, 21423718, 29354284.

Response: Thank you for bringing these references to our attention. We have added these citations to the revised manuscript.

8. The authors integrated their cassettes into what they describe as “the well-expressed Flp-In locus”. I believe that the term “Flp-In locus” is not accurate. These are cell lines that have been derived by inserting the FRT site in random locations, typically then selecting clones that are transcriptionally active and have low epigenetic silencing.

Response: Thank you for this comment! We dropped “Flp-In locus” and instead referred to it as the “genomic FRT site”, which is identical between the mNF and mPF cells.

9. Did the authors start with a monoclonal Flp-In cell line? What is the actual location of integration? Were the circuits integrated as a single copy? The authors have to verify the location and copies.

Response: We started with a monoclonal Flp-In cell line which was purchased from Invitrogen. We tried, but we do not think it is practical to determine the location of the genomic FRT site where the gene circuits are integrated. Instead, we accomplished what was more feasible, specifically to verify that the gene circuits are integrated as a single copy with a promoter trap integrant cell enrichment scheme. Thus, we determined the copy numbers for each gene circuit in the CHO cell lines with qPCR, and since CHO cells have two copies of Vinculin¹⁸, the relative copy number for each gene circuit (one copy of *PuroR* for every two Vinculin copy ~ 0.5) supports single-copy integration (**Supplementary Fig. S8**). Considering the clonal parental cells and selection with Hygromycin B that requires integration of the start-codon-less HygroB resistance gene into the genomic FRT site containing a start codon as a promoter trap, we conclude that the gene circuits are located in single copies at the same unique genomic FRT locus.

References for the Response to Reviewers

- 1 Weinberger, L. S., Dar, R. D. & Simpson, M. L. Transient-mediated fate determination in a transcriptional circuit of HIV. *Nature Genetics* **40**, 466-470, doi:10.1038/ng.116 (2008).
- 2 Isackson, P. J. & Bertrand, K. P. Dominant negative mutations in the Tn10 Tet repressor - evidence for use of the conserved helix-turn-helix motif in DNA-binding. *Proceedings of the National Academy of Sciences of the United States of America* **82**, 6226-6230, doi:DOI 10.1073/pnas.82.18.6226 (1985).
- 3 Nevozhay, D., Adams, R. M., Murphy, K. F., Josic, K. & Balázsi, G. Negative autoregulation linearizes the dose-response and suppresses the heterogeneity of gene expression. *Proceedings of the National Academy of Sciences of the United States of America* **106**, 5123–, doi:10.1073/pnas.0809901106 (2009).
- 4 Gupta, R. S. & Siminovitch, L. Genetic and biochemical studies with the adenosine analogs toyocamycin and tubercidin: mutation at the adenosine kinase locus in Chinese hamster cells. *Somatic Cell Genetics* **4**, 715-735 (1978).
- 5 Bradley, W. E. & Letovanec, D. High-frequency nonrandom mutational event at the adenine phosphoribosyltransferase (aprt) locus of sib-selected CHO variants heterozygous for aprt. *Somatic Cell Genetics* **8**, 51-66 (1982).
- 6 Rabin, M. S. & Gottesman, M. M. High frequency of mutation to tubercidin resistance in CHO cells. *Somatic Cell Genetics* **5**, 571-583 (1979).
- 7 Zhang, S. *et al.* Mutation Detection in an Antibody-Producing Chinese Hamster Ovary Cell Line by Targeted RNA Sequencing. *Biomed Res Int* **2016**, 8356435, doi:10.1155/2016/8356435 (2016).
- 8 Acar, M., Becskei, A. & van Oudenaarden, A. Enhancement of cellular memory by reducing stochastic transitions. *Nature* **435**, 228-232, doi:10.1038/nature03524 (2005).
- 9 Acar, M., Mettetal, J. T. & van Oudenaarden, A. Stochastic switching as a survival strategy in fluctuating environments. *Nature Genetics* **40**, 471-475, doi:10.1038/ng.110 (2008).
- 10 Balaban, N. Q. Persistence: mechanisms for triggering and enhancing phenotypic variability. *Curr Opin Genet Dev* **21**, 768-775, doi:10.1016/j.gde.2011.10.001 (2011).
- 11 Nevozhay, D., Adams, R. M., Van Itallie, E., Bennett, M. R. & Balázsi, G. Mapping the environmental fitness landscape of a synthetic gene circuit. *PLOS Computational Biology* **8**, e1002480, doi:10.1371/journal.pcbi.1002480 (2012).
- 12 Brock, A., Chang, H. & Huang, S. Non-genetic heterogeneity—a mutation-independent driving force for the somatic evolution of tumours. *Nature reviews. Genetics* **10**, 336–342, doi:10.1038/nrg2556 (2009).
- 13 Levin-Reisman, I. *et al.* Antibiotic tolerance facilitates the evolution of resistance. *Science* **355**, 826-830, doi:10.1126/science.aaj2191 (2017).
- 14 Coward, J. & Harding, A. Size does matter: Why polyploid tumor cells are critical drug targets in the War on Cancer. *Frontiers in Oncology* **4**, 123, doi:10.3389/fonc.2014.00123 (2014).
- 15 Charlebois, D. A., Balazsi, G. & Kaern, M. Coherent feedforward transcriptional regulatory motifs enhance drug resistance. *Phys Rev E Stat Nonlin Soft Matter Phys* **89**, 052708, doi:10.1103/PhysRevE.89.052708 (2014).
- 16 Charlebois, D. A., Abdennur, N. & Kaern, M. Gene expression noise facilitates adaptation and drug resistance independently of mutation. *Physical Review Letters* **107**, doi:10.1103/PhysRevLett.107.218101 (2011).
- 17 Dehairs, J., Talebi, A., Cherifi, Y. & Swinnen, J. V. CRISP-ID: decoding CRISPR mediated indels by Sanger sequencing. *Scientific Reports* **6**, 28973, doi:10.1038/srep28973 (2016).

- 18 Xu, X. *et al.* The genomic sequence of the Chinese hamster ovary (CHO)-K1 cell line. *Nature Biotechnology* **29**, 735-U131, doi:10.1038/nbt.1932 (2011).
- 19 Tang W, *et al.* Faithful expression of multiple proteins via 2A-peptide self-processing: a versatile and reliable method for manipulating brain circuits. *J Neurosci.* **29**(27), 8621-9, doi: 10.1523/JNEUROSCI.0359-09.2009. (2009).
- 20 Papapetrou E, *et al.* Stoichiometric and temporal requirements of Oct4, Sox2, Klf4, and c-Myc expression for efficient human iPSC induction and differentiation. *Proceedings of the National Academy of Sciences of the United States of America* **106**(31), Proceedings of the National Academy of Sciences of the United States of America 106, 12759-64, doi: 10.1073/pnas.0904825106 (2009).

REVIEWERS' COMMENTS:

Reviewer #1 (Remarks to the Author):

The revised study is strong and the authors conclusions are supported by evidence to such a degree that doubting them would be unreasonable. The authors have appropriately addressed the concerns of the reviewers whenever possible and performed an incredible range of validation experiments. Investigations like the ones completed by the authors are complicated because so many variables can influence adaptation to cytotoxicity and the development of drug resistance. I am very pleased that the authors conducted many of the experiments suggested by the reviewers and believe that any outstanding issues are far less important than (finally) making the study available to the community.

The most significant contribution, to me, is that the study provides direct supports for the hypothesis articulated by Amy Brock and Sui Huang a decade ago in Nature Reviews Genetics about the possibility that cell-to-cell variability in gene expression, the time-scale of gene expression and acquired mutations may contribute to the development of drug resistance of tumors. Although numerous studies using simulations, particularly a study by one of the contributing authors in 2011, and non-mammalian model systems collectively have supported the hypothesis as valid to varying degree, I believe that the authors in the present study has demonstrated beyond reasonable doubt validates the Brock hypothesis.

It puzzles me, however, is that the authors on one hand recognizes the potentially profound impact of their findings for developing a deeper understanding of drug cancer resistance, but on the other hand fail to emphasize the significance of their findings in the context of the Brock hypothesis (Ref. 15) and its subsequent formal theoretical validation (Ref. 11). I really hope the authors will have an opportunity to better communicate their work prior to publication and replace the emphasis on networks with one that speaks more directly to the biological and biomedical implications of their findings.

Reviewer #2 (Remarks to the Author):

The authors have addressed all of our concerns. A few comments (begin with *).

* What are normalized mean and normalized CV? How do they differ from raw CV I didn't see a definition.

(1) It seems impossible to differentiate the effect of noise from the different in network architecture. All of the effects may be due to the positive feedback loop, and not the high noise.

* Good enough for us. It's not essential to separate network from noise in this paper. However, we think there are still quite a few remnants of the 'noise is causal' model. The authors can suggest, but not claim, that noise is causal for the difference. For example, change "Testing the role of network noise in mammalian cell evolution as in yeast requires a similar control feat." -> "Testing the role of network architecture and noise..." and "Accordingly, the low-noise network could maintain resistance" -> "Accordingly, the network with negative feedback and low noise could maintain resistance", or something similar.

(A) High expressing cells with the PF circuit may maintain high expression, and this could generate the increased survival at high drug concentration.

* Ok, the key point here is that there is no temporal overlap between the emerging of drug-

resistant colonies under high concentrations of the drug (~few weeks) and the relaxation of high gene expression in mPF cells (~few days)? Add this point to the manuscript.

"if slow fluctuations would underlie drug resistance without any phenotypic switching to a cell cycle arrested persister-state, the surviving cells should start visibly growing in a few days, as opposed to the experimentally observed multi-week delays without growth." Is an assumption. Likely, but an assumption.

It is generally difficult to compare expression values across figures. Make it more obvious that 4b is a zoom of 4a.

Response: Thank you for this suggestion - we have now added a black bracket in Fig. 4a that represents the expression range in Fig. 4b.

* Why on the panel A -- CV for mPF is ~0.7 and on the panel B -- ~0.9-1. Why the mismatch?

The figures, and axis labels, are generally difficult to understand.

Response: We have improved the figures and axis labels to address this concern.

* Fig 6 (panels B - F -- please highlight better differences between lines, it is very hard to distinguish lines between P and G type of cells); If you want to keep the color scheme, one could be solid.

Fig 7 -- please add axis labels and titles to each panel

S19 -- it seems to me that it is simply a part of figure 7, is it true? If not, what is different?

Reviewer #3 (Remarks to the Author):

Thank you for addressing my concerns. I have no further comments.

REVIEWERS' COMMENTS:

Reviewer #1 (Remarks to the Author):

The revised study is strong and the authors conclusions are supported by evidence to such a degree that doubting them would be unreasonable. The authors have appropriately addressed the concerns of the reviewers whenever possible and performed an incredible range of validation experiments. Investigations like the ones completed by the authors are complicated because so many variables can influence adaptation to cytotoxicity and the development of drug resistance. I am very pleased that the authors conducted many of the experiments suggested by the reviewers and believe that any outstanding issues are far less important than (finally) making the study available to the community.

Response: Thank you for these encouraging comments and for the appreciation of our work.

The most significant contribution, to me, is that the study provides direct supports for the hypothesis articulated by Amy Brock and Sui Huang a decade ago in Nature Reviews Genetics about the possibility that cell-to-cell variability in gene expression, the time-scale of gene expression and acquired mutations may contribute to the development of drug resistance of tumors. Although numerous studies using simulations, particularly a study by one of the contributing authors in 2011, and non-mammalian model systems collectively have supported the hypothesis as valid to varying degree, I believe that the authors in the present study has demonstrated beyond reasonable doubt validates the Brock hypothesis.

Response: Thank you for pointing this out. We agree.

It puzzles me, however, is that the authors on one hand recognizes the potentially profound impact of their findings for developing a deeper understanding of drug cancer resistance, but on the other hand fail to emphasize the significance of their findings in the context of the Brock hypothesis (Ref. 15) and its subsequent formal theoretical validation (Ref. 11). I really hope the authors will have an opportunity to better communicate their work prior to publication and replace the emphasis on networks with one that speaks more directly to the biological and biomedical implications of their findings.

Response: We have made efforts to better emphasize how our findings relate to these earlier pioneering papers. Specifically, we have added a paragraph to the Discussion regarding this connection.

Reviewer #2

The authors have addressed all of our concerns. A few comments (begin with *).

Response: Thank you for appreciating our efforts to address your constructive comments, which really helped us improve the manuscript.

* What are normalized mean and normalized CV? How do they differ from raw CV I didn't see a definition.

Response: Thank you for pointing out the need to clarify this part of data processing. We now more clearly describe the normalization and processing of flow cytometry data in the Methods section and briefly describe it in the Results of the revised manuscript.

We use low and high fluorescence controls to normalize the raw flow cytometry readings (data) and to minimize technical measurement variation between experiments. The average fluorescence of parental CHO-Flp cells, which exhibit auto-fluorescence but lack any gene circuits serve as the low fluorescence control $EGFP_{\text{auto}}$. The average fluorescence of the highest peak from calibration beads (BD, 653144) with 8 distinct fluorescent peaks serve as the high fluorescence control $EGFP_{\text{max}}$.

We first filter out non-expressing, circuit-silenced cells from the raw data:

$$EGFP_{\text{filtered}}^i = EGFP_{\text{raw}}^i \mid (EGFP_{\text{raw}}^i > EGFP_{\text{cutoff}})$$

where $EGFP_{\text{filtered}}^i$ is the fluorescence for individual cells (labeled by the index i) that have an unsilenced circuit, $EGFP_{\text{raw}}^i$ is the raw fluorescence from any cell (i) with the gene circuit, and $EGFP_{\text{cutoff}}$ is a fluorescence threshold selected above auto-fluorescence but below uninduced, basal circuit expression:

$$EGFP_{\text{auto}} < EGFP_{\text{cutoff}} < EGFP_{\text{basal}}$$

The $EGFP_{\text{cutoff}}$ (2e3 a.u.) was shown to threshold non-fluorescent cells from cells with uninduced circuits based on the mean and standard deviation of auto-fluorescence of parental cells not exceeding the threshold until expression is three standard deviations from the auto-fluorescence mean. Additionally, the distribution non-expressing peak and basal expression peak has a local minimum that aligns with the $EGFP_{\text{cutoff}}$. The filter is justified because cells with a completely silenced circuit display no evidence of circuit re-activation (see **Supplementary Fig. S36**).

We then normalize fluorescence for individual cells that have an unsilenced circuit by the following equation:

$$EGFP_{\text{norm}}^i = \frac{EGFP_{\text{filtered}}^i - EGFP_{\text{auto}}}{EGFP_{\text{max}}}$$

The fact that individual bead distributions align best after the above calculations validates this normalization method.

Using each individual normalized fluorescence reading, we calculated the normalized mean and CV directly from the $EGFP_{\text{norm}}^i$ values using standard formulas: $CV = \sigma(EGFP_{\text{norm}}^i) / \mu(EGFP_{\text{norm}}^i)$ where

$$\mu(EGFP_{norm}^i) = \frac{1}{N} \sum_1^N EGFP_{norm}^i \quad \text{and} \quad \sigma^2(EGFP_{norm}^i) = \frac{1}{N} \sum_1^N (EGFP_{norm}^i)^2.$$

We calculated the raw mean and CV directly from the raw data, without any filtering or normalization. We verified that the normalized CV approximates well the raw CV (see **Figs. 2b, 3b, 4a** vs. **Supplementary Figure 2b**; and **Fig. 4b, Supplementary Figure 9b** vs. **Supplementary Figure 9a**). On the other hand, the normalized population mean usually differs from the corresponding raw mean (see **Figs. 2a, 3a, 4a** vs. **Supplementary Figure 2a**; **Fig. 4b, Supplementary Figure 9b** vs. **Supplementary Figure 9a**; and **Figs. 7b-d** vs. **Supplementary Figure 20**).

(1) It seems impossible to differentiate the effect of noise from the different in network architecture. All of the effects may be due to the positive feedback loop, and not the high noise.

* Good enough for us. It's not essential to separate network from noise in this paper. However, we think there are still quite a few remnants of the 'noise is causal' model. The authors can suggest, but not claim, that noise is causal for the difference. For example, change "Testing the role of network noise in mammalian cell evolution as in yeast requires a similar control feat." -> "Testing the role of network architecture and noise..." and "Accordingly, the low-noise network could maintain resistance" -> "Accordingly, the network with negative feedback and low noise could maintain resistance", or something similar.

Response: Thank you for pointing this out. We have tried to further diminish direct implications of noise as the source of the differences in drug resistance evolution as much as possible.

(A) High expressing cells with the PF circuit may maintain high expression, and this could generate the increased survival at high drug concentration.

* Ok, the key point here is that there is no temporal overlap between the emerging of drug-resistant colonies under high concentrations of the drug (~few weeks) and the relaxation of high gene expression in mPF cells (~few days)? Add this point to the manuscript.

Response: Thank you for bringing this up. We now explicitly mention the lack of temporal overlap between memory and adaptation times at the end of the results section describing the drug treatment growth curves.

"if slow fluctuations would underlie drug resistance without any phenotypic switching to a cell cycle arrested persister-state, the surviving cells should start visibly growing in a few days, as opposed to the experimentally observed multi-week delays without growth." Is an assumption. Likely, but an assumption.

Response: We agree - without an experiment comparing phenotypic switching to a persister state vs. no switching, it is an assumption supported by the computational model, according to which the long delay (adaptation time) before quick regrowth is inconsistent with slower and larger PF fluctuations alone facilitating adaptation compared to NF. We will clarify that this is an assumption in the text. "If slow fluctuations would underlie drug resistance without any phenotypic switching to other cell states, the computational model assumingly suggests slow but visible growth in a few days, as opposed to the experimentally observed multi-week delays without growth."

It is generally difficult to compare expression values across figures. Make it more obvious that 4b is a zoom of 4a.

Response: Thank you for this suggestion - we have now added a black bracket in Fig. 4a that represents the expression range in Fig. 4b.

* Why on the panel A -- CV for mPF is ~0.7 and on the panel B -- ~0.9-1. Why the mismatch?

Response: We process the flow cytometry data in two different ways. In the main text, we filter out non-expressing cells and use low-expressing and high-expressing controls to normalize the data before calculating the CV and the mean. In the Supporting Information we use the raw data directly, without any filtering and normalization for CV and mean calculations. Accidentally, the non-expressing subpopulation was not filtered out in **Figs. 2b,c; 3b,c; 4a,b; 7b-d; Supplementary Figures 2c-f; 4b-e; 6b-e; 9b** before the normalized CV and mean calculations. We have now corrected the figure by filtering and normalizing the data as described in the Methods section. Although the difference in CV tightened, a difference of 15-25% between the mPF comprehensive dose response (**Fig. 4a**) and pre-treatment (**Fig. 4b; Supplementary Figure 9b**) measurements still remains. This is normal experimental variability, which occurs when working with mammalian cell lines. Also, the filtered and normalized means for experiment set 2 (35 and 50 $\mu\text{g}/\text{mL}$ Puromycin treatment) now differ by 8% compared to the previous 5%, while still remaining statistically significant. Thus, now we define a decoupled noise point with the means differing less than 10%. This does not change any conclusions, since the mPF cells still have a lower mean than the mNF cells at pre-treatment, which only strengthens the argument that high noise and positive feedback networks facilitate drug resistance compared to low noise and negative feedback networks.

The figures, and axis labels, are generally difficult to understand.

Response: We have improved the figures and axis labels to address this concern.

* Fig 6 (panels B - F -- please highlight better differences between lines, it is very hard to distinguish lines between P and G type of cells); If you want to keep the color scheme, one could be solid.

Response: We have made one of the similarly colored lines solid instead of dashed.

Fig 7 -- please add axis labels and titles to each panel

Response: We have added axis labels and titles to the unlabeled panels in Figure 7.

S19 -- it seems to me that it is simply a part of figure 7, is it true? If not, what is different?

Response: Thank you for pointing this out. Figure S19a was mistakenly duplicated from Figure 7. Figure S19 displays the raw flow cytometry data without normalization. The normalized and raw CV is very similar, as described above. To be consistent, we display the normalized and raw fluorescent values for

pre-treatment (**Fig. 4a,b; Supplementary Figure 9**) and after treatment (**Fig. 7b-d; Supplementary Figure 20**).